# A Survey on Over-smoothing and Over-squashing: Unified Propagation Perspectives on Graph Neural Networks and Transformers

**Álvaro Arroyo** *University of Oxford*                                         *alvaro.arroyo@eng.ox.ac.uk*

**Federico Barbero** *University of Oxford*

**Hugh Blayney** *University of Oxford*

**Michael Bronstein** *University of Oxford, AITHYRA*

**Xiaowen Dong** *University of Oxford*

**Pietro Liò** *University of Cambridge*

**Razvan Pascanu** *Mila*

**Pierre Vandergheynst** *EPFL*

**Reviewed on OpenReview:** *https://openreview.net/forum?id=H9zhC5pVnH*

## Abstract

Decoder-Transformers have achieved remarkable success and have laid the groundwork for the development of Large Language Models (LLMs). At the core of these models is the *self-attention matrix*, which allows different tokens to interact with each other. This process is remarkably similar to the message-passing mechanism used in Graph Neural Networks (GNNs), and as such decoder-Transformers suffer many of the optimization difficulties studied extensively in the GNN literature. In this paper, we present a unified graph perspective that bridges the theoretical understanding of decoder-Transformers and GNNs. We systematically examine how well-known phenomena in GNNs, such as **over-smoothing** and **over-squashing**, directly manifest as analogous issues like **rank collapse** and **representational collapse** in deep Transformer architectures. By interpreting Transformers' self-attention as a learned adjacency operator, we reveal shared underlying principles governing signal propagation and demonstrate how insights from one field can illuminate challenges and solutions in the other. We analyze the role of architectural components like residual connections, normalization, and causal masking in these issues. We aim to provide a framework for understanding how information flows through deep learning models that perform sequence *mixing* through an adjacency operator, and to highlight areas for cross-pollination of research, as well as to provide a comprehensive reference for researchers interested in the underpinnings of these architectures.

## 1 Introduction

The remarkable success of large language models (LLMs) in natural language processing, code generation, and multimodal reasoning has spurred intensive research into understanding their internal mechanisms and inductive biases. At the heart of modern LLMs lies the Transformer architecture (Vaswani et al., 2017), which leverages self-attention to capture both short and long-range dependencies in sequences. Despite their empirical success, little is understood about their learning dynamics, how their expressive power scales with depth and width (sequence length), and how to diagnose and mitigate different failure modes. In parallel, the

field of graph neural networks (GNNs) has matured into a rich intersection of spectral graph theory, dynamical systems, and message-passing paradigms (Gilmer et al., 2017; Defferrard et al., 2016; Kipf & Welling, 2017). GNNs process relational data by propagating node features along graph edges. While these two architectures seem distinct, it has been argued (Joshi, 2025; Pappone, 2025) that Transformers are an instance of a GNN, which is due to the analogous way that tokens (or nodes) are mixed and iteratively updated through successive layers of the network. This similarity in information processing suggests that many insights and optimization-related problems from GNN theory, such as over-smoothing (Oono & Suzuki, 2020) and over-squashing (Topping et al., 2021), may shed light on analogous phenomena in decoder-Transformers, and hence, LLMs.

In this perspective and survey paper, we advocate for a *graph approach* to understanding LLMs, one that unifies sequence and graph modeling under the common umbrella of **mixing architectures**. We organize our exposition around the core idea of **information propagation**. In particular, we elaborate on how the normalized adjacency matrix – the key component of mixing architectures – affects information flow in *depth* (in the number of layers) and *width* (in the diameter of the graph or sequence). We examine how concepts like **over-smoothing** and **over-squashing**, well-studied in GNNs, have manifested as analogous challenges named **rank collapse** and **representational collapse** in Transformer literature. We focus specifically on decoder-style causal Transformers as these are both widely used in practice and studied in recent literature, and the causal masking makes propagation effects particularly relevant. However, we highlight that the core concept of studying mixing via the adjacency operator extends to other Transformer families and modalities such as images (Dosovitskiy et al., 2021), audio (Gong et al., 2021) and point clouds (Zhao et al., 2021); we refer readers to Islam et al. (2024) for a comprehensive survey of applications. By drawing these connections and citing the most relevant work in both bodies of literature, we aim to provide the reader with a cohesive perspective on the shared underlying principles governing the behavior of both GNNs and decoder-Transformers, highlighting how advancements in one domain can inform theoretical understanding and practical improvements in the other.

**Contributions and Scope**   This paper makes several contributions:

1. (§4.1) We formalize decoder self-attention as a learned adjacency (mixing) operator and relate it to message-passing GNN updates, making explicit the shared structure of a mixing step followed by a feature update.

2. (§3–§4.2) We organize depth- and width-related pathologies under a single information-propagation lens, connecting over-smoothing/vanishing gradients and over-squashing (present in the GNN literature) to rank/representational collapse and long-context degradation mechanisms (present in the Transformer literature).

3. (§4.3) We systematize mitigation strategies as mechanisms that preserve representation diversity and stable gradient flow, and compare how these themes appear across both GNNs and Transformer variants.

4. We present a propagation-centric framework intended to encourage cross-pollination between the GNN and Transformer communities, highlighting transferable tools from graph signal processing and network science for attention-based models, and vice versa.

## 2   Background and Foundations

In this section, we aim to provide the reader with the relevant background in both sequence modeling and GNNs. We begin by introducing foundational sequence-modeling architectures – starting with Recurrent Neural Networks and their gradient-flow challenges, progressing to the attention mechanism, and culminating in the Transformer architecture. We then provide an overview of graph neural networks and their message-passing foundations, as well as their associated challenges.

### 2.1   Background on Sequence Modeling

Sequence modeling is a foundational concept in machine learning that deals with causally ordered data, such as text (Achiam et al., 2023), speech (Goel et al., 2022), or time series (Lim & Zohren, 2021). While such sequential tasks have often relied on handcrafted priors or simple linear models with closed-form solutions,

the increased availability of compute power and data has led to the advent of deep learning approaches. Here, we cover some of the most common frameworks, providing a foundation for gradient-based phenomena that underpin later sections.

### 2.1.1   Fundamentals of Sequence Modeling

Deep neural networks build intermediate representations by applying a series of iterated nonlinear transformations. This process enables the progressive abstraction of raw input data into higher-level features. Many architectures in this domain can be viewed as sequence-to-sequence mappings,

$$\mathbf{g_\theta} : \mathbf{x}^{(0:t)} \mapsto \mathbf{y}^{(0:t)}, \tag{1}$$

where the input and output sequences up to time step $t$ are defined as

$$\begin{aligned} \mathbf{x}^{(0:t)} &= (\mathbf{x}^{(0)}, \mathbf{x}^{(1)}, \dots, \mathbf{x}^{(t)}), \\ \mathbf{y}^{(0:t)} &= (\mathbf{y}^{(0)}, \mathbf{y}^{(1)}, \dots, \mathbf{y}^{(t)}). \end{aligned} \tag{2}$$

Here, each input vector $\mathbf{x}^{(k)} \in \mathbb{R}^d$ represents the data at time step $k$, while each output vector $\mathbf{y}^{(k)} \in \mathbb{R}^m$ corresponds to the model's prediction at that time step. The parameter set $\boldsymbol{\theta}$ corresponds to all learnable weights of the model.

In sequence modeling, the inherent temporal (or causal) structure of the data is captured by having the *hidden state* $\mathbf{h}^{(t)}$ be a function of all previous information from the input sequence:

$$\mathbf{h}^{(t)} = \mathbf{g}_{\boldsymbol{\theta}}^{\mathrm{enc}}\left(\mathbf{x}^{(0:t)}\right) \tag{3}$$

where $\mathbf{x}^{(0:t)}$ denotes the collection of all inputs from time 0 to $t$, and $\mathbf{g}_{\boldsymbol{\theta}}^{\mathrm{enc}}$ is a learnable encoder function. The output at time step $t$ is then generated by the decoder,

$$\mathbf{y}^{(t)} = \mathbf{g}_{\boldsymbol{\theta}}^{\mathrm{dec}}\left(\mathbf{h}^{(0:t)}\right), \tag{4}$$

with $\mathbf{g}_{\boldsymbol{\theta}}^{\mathrm{dec}}$ representing the learnable decoder function.

### 2.1.2   The Vanishing and Exploding Gradient Problem

A particularly important instantiation of the encoder $\mathbf{g}_{\boldsymbol{\theta}}^{\mathrm{enc}}$ is obtained by imposing a *recurrent* structure on the hidden state. In Recurrent Neural Networks (RNNs), the hidden representation is updated recursively as

$$\mathbf{h}^{(t)} = F_{\boldsymbol{\theta}}\left(\mathbf{h}^{(t-1)}, \mathbf{x}^{(t)}\right), \tag{5}$$

for some parametric transition map $F_{\boldsymbol{\theta}}$. When unfolded over $T$ time steps, this recursion defines a depth-$T$ computational graph with shared parameters across layers. Training such models via backpropagation through time (BPTT) therefore requires propagating gradients through many repeated applications of the same state-transition Jacobian.

This recursive depth-with-weight-sharing structure, which first exposed the vanishing and exploding gradient problem in sequence modeling (Bengio, 1994; Hochreiter & Schmidhuber, 1997; Pascanu et al., 2013). This occurs because, during BPTT, the gradients of the hidden states are computed by repeatedly applying the chain rule across many time steps. As a result, the process involves multiplying a long sequence of Jacobians that are correlated to each other due to being the Jacobian of the same update rule. When this sequence is long, the accumulated product can either shrink rapidly toward zero or blow up uncontrollably, which leads to uncontrolled gradient propagation. In particular, following Pascanu et al. (2013), we let $\mathcal{L}^{(t)}$ denote the loss at time step $t$. The total loss over a sequence of length $T$ is then given by

$$\mathcal{L} = \sum_{t=1}^{T} \mathcal{L}^{(t)}. \tag{6}$$

Accordingly, the gradient of the total loss with respect to the parameters $\boldsymbol{\theta}$ can be expressed as

$$\frac{\partial \mathcal{L}}{\partial \boldsymbol{\theta}} = \sum_{t=1}^{T} \sum_{k=1}^{t} \left( \frac{\partial \mathcal{L}^{(t)}}{\partial \mathbf{h}^{(t)}} \cdot \frac{\partial \mathbf{h}^{(t)}}{\partial \mathbf{h}^{(k)}} \cdot \frac{\partial \mathbf{h}^{(k)}}{\partial \boldsymbol{\theta}} \right). \tag{7}$$

As identified by Pascanu et al. (2013), a major issue arises from the product Jacobian,

$$\mathbf{J} = \frac{\partial \mathbf{h}^{(t)}}{\partial \mathbf{h}^{(k)}} = \prod_{i=k+1}^{t} \frac{\partial \mathbf{h}^{(i)}}{\partial \mathbf{h}^{(i-1)}} = \prod_{i=k+1}^{t} \mathbf{J}_i. \tag{8}$$

If $\|\mathbf{J}_k\|_2 \approx \lambda$ for all layers, then

$$\|\mathbf{J}\|_2 \leq \lambda^{t-k}. \tag{9}$$

Thus, it is necessary that $\lambda \approx 1$ for gradients to neither vanish nor explode—a condition often referred to as the *edge of chaos*. Specifically, if $\lambda < 1$ for certain terms in the gradient sum, these terms will disappear and will be absent from the gradient calculation. Conversely, if $\lambda > 1$, these gradients will explode and the gradient norm will increase greatly. RNN architectures such as LSTMs (Hochreiter & Schmidhuber, 1997) or GRUs (Cho et al., 2014) mitigate vanishing gradients via gated additive recurrence, but remain susceptible to exploding gradients and also struggle with long sequences. As such, a very relevant line of research has been dedicated to fixing this issue, examples of which include norm-preserving weight constraints (Arjovsky et al., 2016; Henaff et al., 2016; Chang et al., 2018; Lezcano-Casado & Martınez-Rubio, 2019), initialization schemes (Tallec & Ollivier, 2018), and physics-inspired inductive biases (Erichson et al., 2020; Keller et al., 2023).

### 2.1.3 Transformers

The introduction of the attention mechanism has led to the Transformer architecture (Vaswani et al., 2017). This model relies on the self-attention mechanism: in a multi-head Transformer, each self-attention head applies scaled dot-product attention to calculate similarity scores between different inputs in the sequence. Let $\mathbf{X} \in \mathbb{R}^{n \times f}$ denote a sequence of $n$ feature vectors, each with dimension $f$. For the $i^{\text{th}}$ head we compute query, key and value matrices using projection matrices $\mathbf{W}_Q^i \in \mathbb{R}^{f \times d_k}, \mathbf{W}_K^i \in \mathbb{R}^{f \times d_k}$ and $\mathbf{W}_V^i \in \mathbb{R}^{f \times d_v}$ as

$$\mathbf{Q}_i = \mathbf{X} \mathbf{W}_Q^i, \qquad \mathbf{K}_i = \mathbf{X} \mathbf{W}_K^i, \qquad \mathbf{V}_i = \mathbf{X} \mathbf{W}_V^i. \tag{10}$$

The scaled dot-product attention is then defined as

$$\mathbf{h}_i = \text{Attention}\left(\mathbf{Q}_i, \mathbf{K}_i, \mathbf{V}_i\right) = \text{softmax}\left(\frac{\mathbf{Q}_i \mathbf{K}_i^T}{\sqrt{d_k}}\right) \mathbf{V}_i, \tag{11}$$

Each head learns an attention function that captures complex dependencies within the input data. With multiple heads, the model jointly attends to different subspaces of the input sequence. The outputs from each head are concatenated to obtain a joint representation:

$$\text{MultiHead}\left(\mathbf{Q}, \mathbf{K}, \mathbf{V}\right) = \text{Concat}(\mathbf{h}_1, \mathbf{h}_2, \ldots, \mathbf{h}_H) \mathbf{W}_O, \tag{12}$$

for output matrix $\mathbf{W}_O \in \mathbb{R}^{Hd_v \times d_o}$. Merging all heads' outputs through a linear function in this way produces a latent representation that encodes the most relevant information from the input features.

In autoregressive tasks, such as next-token prediction, a causal mask is essential to prevent a token from accessing future information. To achieve this, the standard attention computation is modified by adding a mask $\mathbf{M}$ to the scaled dot-products, as shown in

$$\text{Attention}(\mathbf{Q}, \mathbf{K}, \mathbf{V}) = \text{softmax}\left(\frac{\mathbf{Q}\mathbf{K}^T}{\sqrt{d_k}} + \mathbf{M}\right) \mathbf{V}, \tag{13}$$

where the mask is defined by

$$M_{ij} = \begin{cases} 0, & j \leq i, \\ -\infty, & j > i, \end{cases} \tag{14}$$

thereby ensuring that positions corresponding to future tokens obtain zero attention after the softmax. This mechanism is critical for maintaining the autoregressive property by ensuring that each token is computed solely on its preceding context, which preserves the sequential nature of the data.

Despite its strengths, self-attention runs in $\mathcal{O}(n^2 d + nd^2)$ time and requires $\mathcal{O}(n^2 + nd)$ memory, for dimension $d$ (assuming $d = f = d_k = d_v$) and sequence length $n$. This arises since forming the score matrix $\mathbf{QK}^T$ costs $\mathcal{O}(n^2 d)$ and the $\mathbf{Q}, \mathbf{K}, \mathbf{V}$ projections cost $\mathcal{O}(nd^2)$. To cope with the computational challenges, these operations are mapped to huge, parallel batched matrix multiplications that GPUs handle with high arithmetic intensity, predictable memory access, and static tensor shapes that allow aggressive kernel fusion. This marriage between algorithm and hardware (also termed the "hardware lottery" (Hooker, 2021)) has led to the effective parameter scaling of attention-based models in practice (Achiam et al., 2023).

## 2.2 Background on Graph Neural Networks

Many real-world systems, from molecular structures and protein–protein interaction networks in biology to citation graphs and social networks in information science, naturally manifest as graphs, where nodes represent entities and edges encode relationships. Learning predictive models on such graph-structured data is crucial for tasks like node classification, link prediction, and graph-level regression, yet traditional machine learning methods that assume independent and identically distributed (i.i.d.) samples cannot exploit the rich relational inductive biases inherent in these domains.

Graph Neural Networks (GNNs) (e.g. Sperduti, 1993; Gori et al., 2005; Scarselli et al., 2008; Bruna et al., 2013; Defferrard et al., 2016) have emerged as a powerful framework for learning on graphs by combining spectral and spatial insights. In the spectral view, graph convolutions are defined via eigendecompositions of a graph shift operator (e.g., the graph Laplacian), enabling filtering in the frequency domain (Bruna et al., 2013; Defferrard et al., 2016); in the spatial view, methods directly aggregate information from each node's local neighborhood. Most contemporary architectures adopt a message-passing paradigm, wherein nodes iteratively exchange and transform "messages" with their neighbors—culminating in the Message-Passing Neural Network (MPNN) framework (Gilmer et al., 2017). Although these two lines of work have been developed independently, it is reasonable to use MPNNs as an umbrella framework to describe both. In this subsection, we will provide the necessary background on MPNNs and discuss related challenges.

### 2.2.1 Message Passing Neural Networks

Let a graph $\mathbf{G}$ be a tuple $(\mathbf{V}, \mathbf{E})$, where $\mathbf{V}$ is the set of nodes and $\mathbf{E}$ is the set of edges. An edge from node $u \in \mathbf{V}$ to node $v \in \mathbf{V}$ is denoted by $(u, v) \in \mathbf{E}$. The graph's connectivity is captured by an adjacency matrix $\mathbf{A} \in \mathbb{R}^{n \times n}$, where $n$ is the number of nodes. We assume that $\mathbf{G}$ is undirected and that each node $v$ is associated with a feature vector $\mathbf{h}_v \in \mathbb{R}^d$. A GNN can be broadly defined as a function

$$\mathbf{f}_{\boldsymbol{\theta}} : (\mathbf{G}, \{\mathbf{h}_v\}) \mapsto \mathbf{y}, \tag{15}$$

with parameters $\boldsymbol{\theta}$ learned via gradient descent, and $\mathbf{y}$ representing a prediction at the node or graph level. These functions typically adopt the message-passing paradigm, which in computing the latent representation at the $k$th-layer can be formulated as:

$$\mathbf{h}_u^{(k)} = \phi^{(k)}\Big(\mathbf{h}_u^{(k-1)}, \psi^{(k)}\big(\{\mathbf{h}_v^{(k-1)} : (u, v) \in \mathbf{E}\}\big)\Big), \tag{16}$$

for $k = 1, \ldots, K$. Here, $\psi^{(k)}$ is a permutation invariant aggregation function, and $\phi^{(k)}$ combines the incoming messages from a node's neighbors with its previous embedding to produce an updated representation. A common aggregation function is given by:

$$\psi^{(k)}\Big(\{\mathbf{h}_v^{(k-1)} : (u, v) \in \mathbf{E}\}\Big) = \sum_{\mathbf{v}} \tilde{\mathbf{A}}_{uv}\, \mathbf{h}_v^{(k-1)}, \tag{17}$$

where $\tilde{\mathbf{A}} = \mathbf{D}^{-\frac{1}{2}} \mathbf{A}\, \mathbf{D}^{-\frac{1}{2}}$ and $\mathbf{D} \in \mathbb{R}^{n \times n}$ is a diagonal matrix with $\mathbf{D}_{ii} = \sum_j \mathbf{A}_{ij}$.

A widely used instance of GNNs is the Graph Convolutional Network (GCN) (Kipf & Welling, 2017), which can be interpreted as an MPNN. Specifically, one can express the node features in matrix form $\mathbf{H}^{(k)} \in \mathbb{R}^{n \times d_k}$, and the GCN update equation is

$$\mathbf{H}^{(k)} = \sigma(\hat{\mathbf{A}} \, \mathbf{H}^{(k-1)} \, \mathbf{W}^{(k-1)}), \tag{18}$$

with

$$\hat{\mathbf{A}} = (\mathbf{D} + \mathbf{I})^{-\frac{1}{2}} (\mathbf{A} + \mathbf{I})(\mathbf{D} + \mathbf{I})^{-\frac{1}{2}}, \tag{19}$$

and $\sigma(\cdot)$ denoting a nonlinearity. Another popular MPNN instance is that of Graph Attention Networks (GATs) (Veličković et al., 2018), where a learned adjacency matrix replaces the fixed normalized adjacency to dynamically modulate connectivity while preserving key spectral properties. In particular, the aggregation is computed as

$$\mathbf{h}_i^{(l+1)} \;=\; \sigma \left( \sum_{j \in \mathcal{N}(i)} \alpha_{ij}^{(l)} \, \mathbf{W} \, \mathbf{h}_j^{(l)} \right), \tag{20}$$

where the attention coefficients are given by

$$\alpha_{ij}^{(l)} \;=\; \frac{\exp\big(\text{LeakyReLU}\big(\mathbf{a}^\top[\mathbf{W}\mathbf{h}_i^{(l)} \,\|\, \mathbf{W}\mathbf{h}_j^{(l)}]\big)\big)}{\sum_{n \in \mathcal{N}(i)} \exp\big(\text{LeakyReLU}\big(\mathbf{a}^\top[\mathbf{W}\mathbf{h}_i^{(l)} \,\|\, \mathbf{W}\mathbf{h}_n^{(l)}]\big)\big)}. \tag{21}$$

Multi-head attention is then introduced by computing $K$ independent sets of coefficients $\{\alpha_{ij}^{(l),k}\}$ (with corresponding weights $\mathbf{W}^{(k)}$, $\mathbf{a}^{(k)}$) and either concatenating or averaging their outputs:

$$\mathbf{h}_i^{(l+1)} = \Big\|_{k=1}^{K} \sigma \left( \sum_{j \in \mathcal{N}(i)} \alpha_{ij}^{(l),k} \, \mathbf{W}^{(k)} \mathbf{h}_j^{(l)} \right) \quad \text{or} \quad \frac{1}{K} \sum_{k=1}^{K} \sigma \left( \sum_{j \in \mathcal{N}(i)} \alpha_{ij}^{(l),k} \, \mathbf{W}^{(k)} \mathbf{h}_j^{(l)} \right). \tag{22}$$

Other notable MPNNs include GraphSAGE (Hamilton et al., 2017), Residual Gated Graph Convolutional Networks (Gated GCNs) (Bresson & Laurent, 2017), and Graph Isomorphism Networks (GIN) (Xu et al., 2018).

**Graph Transformers** We note that a large body of work studies Graph Transformers, i.e., Transformer-style architectures designed for graph-structured inputs. These methods typically adapt self-attention to operate over nodes by (i) treating nodes as tokens, (ii) restricting or biasing attention using the graph structure (e.g., via adjacency-based sparsification or structural biases), and (iii) injecting graph positional/structural encodings (e.g., shortest-path distances, centrality, Laplacian features) so that attention is aware of topology rather than purely content-based. Many Graph Transformers are motivated by limitations of classical message-passing GNNs and by the desire to combine local neighborhood aggregation with global token mixing in a controllable and scalable way.

Representative architectures include Graphormer, which augments attention with graph structural biases such as shortest-path and centrality encodings (Ying et al., 2021), GraphGPS, which proposes a practical "recipe" combining local message passing with global attention while retaining linear complexity in nodes/edges (Rampášek et al., 2022), and approaches that study graph inductive biases in Transformers without explicit message passing (Ma et al., 2023). We also highlight recent surveys that provide a systematic overview of Graph Transformer design choices (Shehzad et al., 2026).[1]

## 3 Signal Propagation in GNNs

In this section, we turn our attention to the mechanisms that govern information propagation in common GNN architectures. In Section 3.1 we introduce well-known issues around stacking GNN layers which lead to the notion of *over-smoothing* used to explain the degradation in performance for common GNN architectures, and in Section 3.2 we introduce the issues of *over-squashing* and long-range communication between distant

---

[1]We emphasize that (graph) Transformers can be viewed as a family of GNNs operating with attention-based message passing; our goal is not to contrast "distinct" model classes, but to use this shared mixing/message-passing viewpoint to build a common language for depth/width pathologies and mitigation strategies across settings.

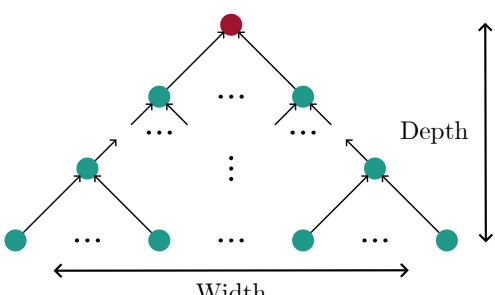

Figure 1: The computational tree for a given root node (in red) for GNNs and Transformers. For GNNs, the children of each node are typically its immediate neighbors in the underlying graph at the previous layer. For Transformers, the children of each node are typically the hidden representations of each (previous, in the case of a causal mask) sequence position at the previous layer. Depth refers to the depth of this tree (number of layers). Width loosely refers to the width of the tree: in GNNs this relates to the node receptive field, and in Transformers to sequence length.

nodes. We make use of the original definitions given to these terms, but also note recent work that highlights the role of vanishing gradients in these concepts (Arroyo et al., 2025) and discuss the interplay between them (Arnaiz-Rodriguez & Errica, 2025). To help unify nomenclature between GNN and Transformer literature, we will separate our discussion into considerations of *depth* (primarily addressed as the issue of over-smoothing in GNN literature) and *width* (addressed as over-squashing in GNN literature). These terms broadly relate to the computational tree, which follows a very similar structure between GNNs and Transformers, thus affording this unification of issues. We visualize an example of such a computational tree in Figure 1. We highlight here a potential nomenclature clash: in this work, "width" does not refer to hidden dimension size but refers approximately to the width of the computational tree, which relates to node receptive field and sequence length in GNNs and Transformers respectively.

## 3.1 The Depth Regime in GNNs: Over-Smoothing and Vanishing Gradients

**Early Approaches: Over-Smoothing.** GNNs are known to severely struggle and break down after the application of a few message-passing layers. Until very recently, this phenomenon has been attributed to the idea of *over-smoothing.* Over-smoothing refers to the phenomenon whereby the repeated application of message-passing operations in GNNs causes the node features to become increasingly similar, visualized in Figure 2. Some early works that analyze this theoretically are Oono & Suzuki (2020); Cai & Wang (2020). A common way to quantify this effect is via the *unnormalized Dirichlet energy*[2] of the node representations, which has been used in a number of works in the area (Rusch et al., 2022). Given a graph $\mathbf{G} = (\mathbf{V}, \mathbf{E})$ with $n$ nodes and node features $\mathbf{H} \in \mathbb{R}^{n \times d}$, the unnormalized Dirichlet energy is defined as

$$\mathcal{E}(\mathbf{H}) = \sum_{(u,v) \in \mathbf{E}} \|\mathbf{h}_u - \mathbf{h}_v\|^2, \tag{23}$$

where $\mathbf{h}_u$ and $\mathbf{h}_v$ denote the feature vectors for nodes $u$ and $v$, respectively.[3] This energy measures the variability between features of adjacent nodes; a lower energy indicates smoother (i.e., more similar) node representations. In many GNN architectures, as more layers are stacked, the iterative aggregation tends to minimize $\mathcal{E}(\mathbf{H})$ to the extent that all node features converge toward a common value, as shown in Figure 3 (left). The common consensus in the literature has been that while some degree of smoothing is beneficial, excessive smoothing leads to a loss of discriminative power, thereby hindering performance in downstream tasks. Several works have aimed to prevent over-smoothing in GNNs, from simpler approaches (Rong et al., 2019; Zhao & Akoglu, 2019) to physics-inspired methods (Chamberlain et al., 2021; Bodnar et al., 2022; Di Giovanni et al., 2022; Bamberger et al., 2024). Other approaches that have emerged over the years

---

[2]Note that other node similarity measures exist, including the normalized Dirichlet energy, used in the original works of (Cai & Wang, 2020) or other measures such as the one employed in (Wu et al., 2024).

[3]Note that the above assumes unitary edge weights.

include Chen et al. (2020); Fang et al. (2023); Lu et al. (2024); Maskey et al. (2024); Roth et al. (2024).

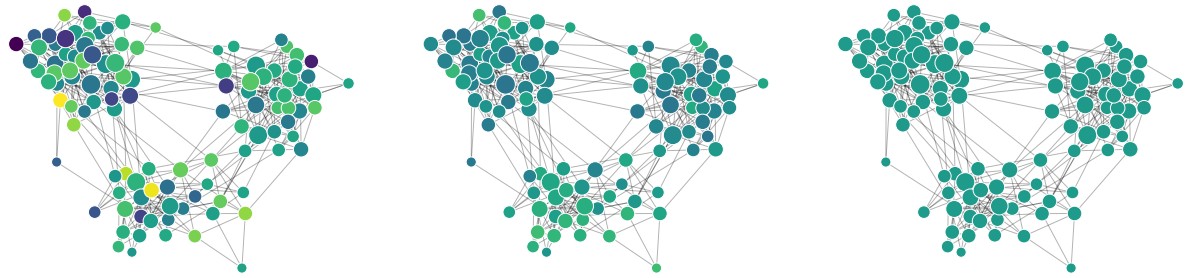

Figure 2: Illustration of the effect of feature collapse on a graph, commonly referred to as over-smoothing. From left to right more GNN layers are applied and node signals, here depicted by color, become more uniform over the graph. Nodes are sized by degree for illustration purposes.

**Issues with Information Propagation and Optimization at Large Depth.** Recently, the degradation in performance of GNNs with a large number of layers has also been associated with fundamental issues of signal propagation and optimization. In particular, Arroyo et al. (2025) highlight the importance of vanishing gradients and the *zero-collapse* phenomenon in GCNs and GATs. The central argument of this work is that while feature collapse is an issue, it should be possible for MPNNs to learn weights that counteract this smoothing effect: however, due to issues of vanishing (and exploding gradients), this is often not the case. This work shows that the same phenomena that gives rise to the over-smoothing effect (the product Jacobian) also gives rise to vanishing gradients. Similar notions were explored in Arnaiz-Rodriguez & Errica (2025), who highlight both that over-smoothing is not an issue in all GNNs, and that over-smoothing does not necessarily cause performance degradation. We note that some of these topics were later explored in Keriven (2025).

Mathematically, the main reason why GCNs and GATs are in expectation more vulnerable to vanishing gradients than standard MLPs is the presence of an additional message-passing step using the normalized adjacency matrix: $\hat{\mathbf{A}}$ in the simplest case of GCN shown in Equation (18). From a dynamical systems perspective, this extra step induces a contraction in the spectrum of the Jacobian, which makes the chain of Jacobians in the backward pass dissipate information at a much higher rate in the backward pass of the network. This is illustrated in Figure 3 (right), together with an example of zero-collapsing of 2-dimensional random projection of node features on the Cora dataset evolved with a GCN (left). To this end, we highlight a number of works that have employed ideas from vanishing gradients or dynamical isometry (Rusch et al., 2022; 2023; Epping et al., 2024; Scholkemper et al., 2024; Wang et al., 2025). Several of these works motivate their methods through comparison to physical systems, which themselves encode stability and favorable optimization properties by design. We also highlight that the divergence effect reported in Arnaiz-Rodriguez & Errica (2025) can be understood from the perspective of optimization. This is due to the fact that the layer-wise Jacobians of some frequently used GNNs (e.g. GIN (Xu et al., 2018) or Gated-GCN (Bresson & Laurent, 2017)) are extremely unstable and will lead to gradient explosion at even moderate depths (see Arroyo et al., 2025).

## 3.2 The Width Regime in GNNs: Over-squashing in Graph Neural Networks

A complementary and highly explored topic in the GNN literature is that of *over-squashing*. Over-squashing was originally introduced in Alon & Yahav (2021) as the compression of information from an exponentially growing receptive field of nodes into a fixed-size vector during the message-passing process. This phenomenon was later linked to a problem of sensitivity and mathematically quantified in Topping et al. (2021); Di Giovanni et al. (2023) through the Jacobian. In particular, this was done by considering an MPNN whose node representations are updated layer-by-layer. Let $\mathbf{h}_v^{(K)}$ denote the representation of node $v$ after $K$ layers, and

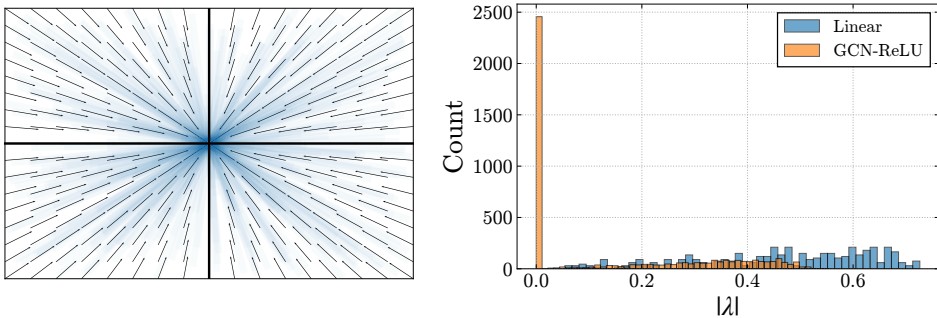

Figure 3: **Left:** 2-Dimensional random projection showing node feature evolution for a GCN on the Cora dataset. **Right:** Histogram of eigenvalue modulus of the Jacobian for linear, linear convolutional, and nonlinear GCN layers. Figures adapted from Arroyo et al. (2025).

$\mathbf{h}_u^{(0)}$ the initial representation of a distant node $u$. In Di Giovanni et al. (2023), the following bound on the sensitivity was derived:

**Theorem 3.2.1** (Sensitivity bounds, (Di Giovanni et al., 2023))**.** *Consider a standard MPNN with $k$ layers, where $c_\sigma$ is the Lipschitz constant of the activation $\sigma$, $w$ is the maximal entry-value over all weight matrices, and $d$ is the embedding dimension. For $u, v \in V$ we have*

$$\left\| \frac{\partial \mathbf{h}_v^{(k)}}{\partial \mathbf{h}_u^{(0)}} \right\| \leq \underbrace{(c_\sigma w d)^k}_{model} \underbrace{(\mathbf{O}^k)_{vu}}_{topology}, \tag{24}$$

*where $\mathbf{O} = c_r \mathbf{I} + c_a \hat{\mathbf{A}} \in \mathbb{R}^{n \times n}$ is the message-passing matrix adopted by the MPNN, and where $c_r$ and $c_a$ are the contributions of the self-connection and aggregation term.*

Notice that the above formulation is composed of both a *model* term and a *topology* term. Many of the earlier works in the space explored rewiring, which explicitly engineered the graph's connectivity to enhance message propagation and mitigate bottlenecks (Gasteiger et al., 2019; Topping et al., 2021; Arnaiz-Rodríguez et al., 2022; Barbero et al., 2023; Gutteridge et al., 2023; Finkelshtein et al., 2024). On the other hand, a parallel strand of research concentrated almost exclusively on the model term itself, crafting non-dissipative update rules or otherwise taming the dynamics without touching the underlying graph (Gravina et al., 2023; 2025; Heilig et al., 2025). However, as pointed out in Arroyo et al. (2025), some of the most robust solutions in the literature combine both perspectives, jointly accounting for the graph topology for information propagation and the non-vanishing node-update dynamics. In particular, performing graph rewiring in isolation does not remove the zero collapse phenomenon and general optimization issues that characterize GNNs, while simply controlling node stability evolution will not enable long-range communication due to the modulation by the inverse square-root of the node degree typically performed when carrying out a message-passing step. We highlight that some of these problems are confined to the use of MPNN architectures, as recent work (Hariri et al., 2025) has also demonstrated the effectiveness of early spectral GNN designs in long-range information propagation. Although the depth terms in this bound mean that over-squashing can be accentuated in deep GNNs, we note too that *shallow* bottlenecks can cause issues; this is of particular relevance when considering long contexts in LLMs (Barbero et al., 2024a). We refer readers to Arnaiz-Rodriguez & Errica (2025) for a discussion on over-squashing and depth, and to Blayney et al. (2025) who investigate over-squashing in shallow GNNs.[4]

## 4 Intersections Between Transformers and GNNs

Having introduced the classical tools and problems in sequence and graph modeling, we are now ready to make connections between the two. Our analysis is based on the interpretation of sequence and Transformer

---

[4]We note that the literature sometimes treats over-squashing as a problem of topological bottlenecks, and sometimes more generally as a problem of bottlenecks in the computational graph: we refer the reader to Arnaiz-Rodriguez & Errica (2025) for a discussion on this and other aspects of over-squashing.

models as message-passing architectures. This intersection can be articulated at several levels. First, at a conceptual level, the core operations of Transformers can be cast as a form of Graph Neural Network (GNN) operating on a fully-connected graph, a perspective originally popularized in Joshi (2025). Second, as a direct consequence of this conceptual similarity, both models suffer from analogous issues in information flow, such as signal degradation in deep models and information bottlenecks. Third, and most importantly for this discussion, the strategies developed to mitigate these issues share a common philosophy: preserving signal integrity and improving information flow through architectural changes like residual connections, normalization, and graph rewiring.

This section will unpack these points. We will first formalize the interpretation of Transformers as GNNs. We will then discuss the shared pathologies related to model depth and input width. Finally, we will dedicate a subsection to a direct comparison of the mitigation strategies that have emerged in both fields, highlighting their parallels.

### 4.1 Interpreting Transformers as GNNs

Even though Transformers and GNNs have traditionally been viewed as distinct architectures[5] , they both interleave MLP blocks (building rich representations) with a "mixing" step that can be interpreted as a message-passing operation. Concretely, in a Transformer layer, we have

$$\mathbf{H}^{(k)} = \mathrm{softmax}\left(\frac{\mathbf{Q}\,\mathbf{K}^{\top}}{\sqrt{d_k}}\right)\mathbf{V} \tag{25}$$

$$= \mathrm{softmax}\left(\frac{\mathbf{H}^{(k-1)}\,\mathbf{W}_Q\left(\mathbf{H}^{(k-1)}\,\mathbf{W}_K\right)^{\top}}{\sqrt{d_k}}\right)\mathbf{H}^{(k-1)}\,\mathbf{W}_V \tag{26}$$

$$= \hat{\mathbf{A}}\,\mathbf{H}^{(k-1)}\,\mathbf{W}_V. \tag{27}$$

where $\hat{\mathbf{A}}$ is the corresponding attention matrix. From this, it is clear that Transformers and GNNs share a similar updating procedure where a learned adjacency matrix, $\hat{\mathbf{A}}$, is used to mix node (token) features. Therefore, MPNNs (especially GCNs and GATs) can be seen as a microcosm for how Transformers operate, and we should expect them to share similar pitfalls and learning dynamics. In particular, the spectral properties of the attention matrix will determine the behavior of the system in depth, while the characteristics of the `softmax` function will determine the properties of the graphs defined by this adjacency matrix. We will elaborate on both of these considerations in the next subsections.

### 4.2 Shared Pathologies: Information Flow Bottlenecks

The structural similarity between GNNs and Transformers leads to analogous failure modes related to information propagation, both in *depth of the model* and *width of the input.*

**The Depth Regime: Over-smoothing and Rank Collapse** Just as deep GNNs suffer from over-smoothing, where node representations converge to a uniform vector, naively stacking Transformer layers leads to rank collapse (Dong et al., 2021). In deep Transformers, the matrix of token embeddings tends to become low-rank, meaning the representations for all tokens collapse into a common subspace and lose their discriminative power. This phenomenon is a direct analogue of over-smoothing: in both cases, the iterative application of a mixing operator ($\hat{\mathbf{A}}$) is theoretically expected to yield increasingly similar node (or token) representations.

The root cause of this shared pathology is the mixing operator $\hat{\mathbf{A}}$ itself, which is the very element that distinguishes these architectures from simple MLPs. This is not merely a representational issue; it has a critical effect on trainability. The same mechanism that drives over-smoothing (in the sense of zero collapse) is also responsible for *vanishing gradients*, making deep GNNs and Transformers notoriously difficult to train (Noci et al., 2022; Arroyo et al., 2025). The mathematical link is the *contractive nature* of the mixing

---

[5]Transformer-based architectures on graphs (often termed Graph Transformers) have been extensively studied and can themselves be viewed as GNNs with learned adjacency operators. Our statement concerns the historical separation of the sequence-Transformer and GNN research communities, which this work seeks to bridge via a unified propagation-centric view.

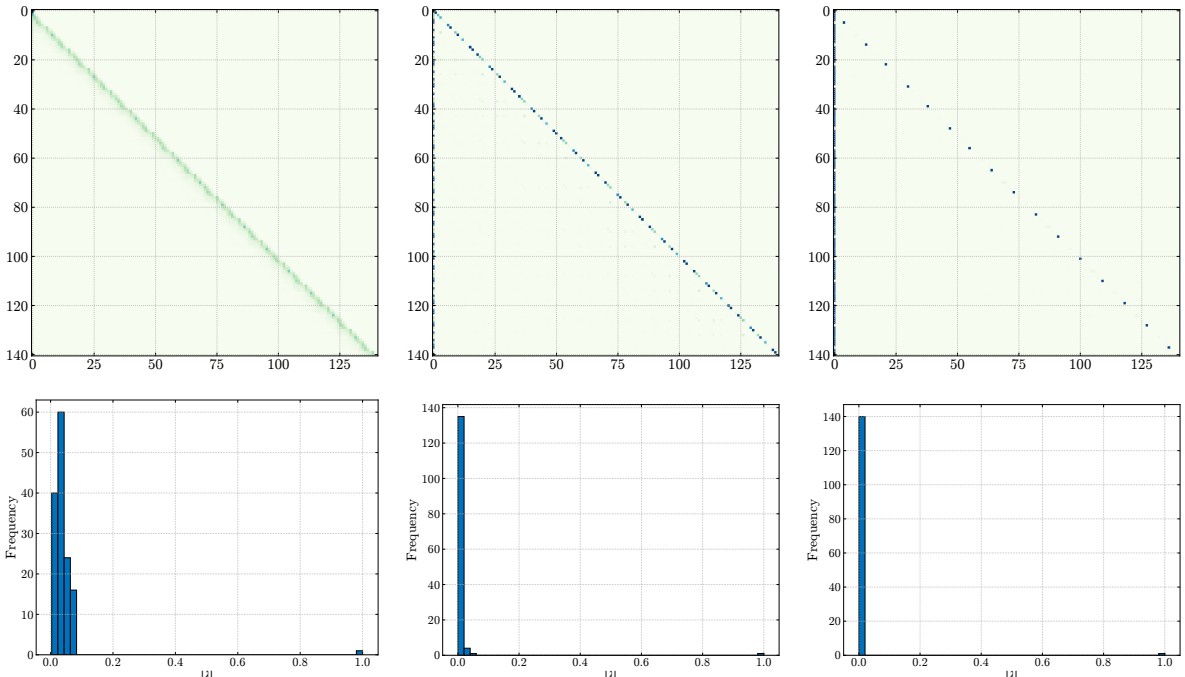

Figure 4: Attention matrices (top) and associated eigenvalue modulus histograms (bottom) for several heads in the early layers of the Gemma 7B model. These histograms demonstrate that the action of the attention matrix is often *highly contractive*

.

matrix's spectrum, as shown in Arroyo et al. (2025). In MPNNs, the use of the normalized adjacency matrix is common, whose eigenspectrum satisfies $|\lambda_i| \leq 1$. In turn, modern LLMs, such as the Llama (Touvron et al., 2023a;b; Grattafiori et al., 2024) and Gemma (Team et al., 2024a;b; 2025) families, impose a triangular (causal) mask for next-token generation. The subsequent application of the `softmax` function additionally means the resulting matrix is *row-stochastic*. This implies it can be analyzed as the transition matrix of a Markov chain, whose eigenvalues are similarly bounded by $|\lambda_i| \leq 1$. Recent work by Wu et al. (2024) has formally extended the notion of rank collapse to these causally masked attention matrices, with complementary random-matrix results in Naderi et al. (2024) employing a spectral approach to analyze their asymptotic behavior.

This creates a fundamental trade-off. On the one hand, contractive mixing causes training instability. On the other hand, empirical evidence from models like Gemma 7B shows that many learned attention heads are, in fact, highly contractive (Figure 4). This suggests the model finds these structures useful for its tasks. This presents a dilemma: to solve its task the model must learn contractive patterns, but these inherently inhibit the gradient flow required for deep, stable training. In practice, this is solved using the architectural solutions discussed in Section 4.3. To make this link precise, we recall two standard consequences of block-wise contraction: repeated composition drives representations to a stable fixed point and monotonically suppresses graph variation (Dirichlet energy), at a rate controlled by the product of layer Lipschitz constants.

**Lemma 4.2.1.** *(Arroyo et al., 2025) Consider a GNN layer $f_K$ as in Equation 18, with non-linearity $\sigma$ such that $\sigma(0) = 0$ (e.g. ReLU or $\tanh$). Then, $f(\mathbf{0}) = \mathbf{0}$, i.e. $\mathbf{0}$ is a fixed point of $f$.*

**Proposition 4.2.2** (Convergence to a unique fixed point (Arroyo et al., 2025).)**.** *Let $\|f_k\|_{Lip} < 1 - \epsilon$ for some $\epsilon > 0$ for all $k = 1 \ldots L$. Then, for $\mathbf{H} \in U \subseteq \mathbb{R}^{nd}$, we have that:*

$$\|f(\mathbf{H})\| < (1 - \epsilon)^K \|\mathbf{H}\| < \|\mathbf{H}\| . \tag{28}$$

*In particular, as $K \to \infty$, $f(\mathbf{H}) \to \mathbf{0}$.*

**Proposition 4.2.3** (Contractions decrease Dirichlet energy. (Arroyo et al., 2025))**.** *Let $f$ be a GNN or Attention-based layer, $|E|$ be the number of edges in $G$, and $\mathbf{H} \in \mathbb{R}^{nd}$. We have the following bound:*

$$\mathcal{E}(f(\mathbf{H})) \leq 2|E| \prod_{k=1}^{K} \|f_k\|_{Lip}^2 \|\mathbf{H}\|^2 . \tag{29}$$

*In particular, if $\|f_k\|_{Lip} < 1 - \epsilon$ for some $\epsilon > 0$ for all $k = 1 \ldots K$, then as $K \to \infty$ we have that $\mathcal{E}(f(\mathbf{H})) \to 0$.*

In particular, these results formalize how a contractive mixing–update block simultaneously (i) collapses node states toward a common fixed point and (ii) decreases Dirichlet energy; in the self-attention setting the same contraction intuition applies in token space, where repeated (masked, row-stochastic) attention mixing can drive the token-embedding matrix toward a low-rank subspace (rank collapse), motivating the need for stabilization mechanisms such as residual pathways and normalization.

**The Width Regime: Over-squashing and Representational Collapse** Beyond the depth-wise issue, decoder-Transformers face "width-wise" issues, which manifest particularly in long-context scenarios. Some of these challenges are analogous to over-squashing in GNNs. In GNNs, the original definition of over-squashing (Alon & Yahav, 2021) stated that this phenomenon occurs when information from an exponentially growing number of nodes at a distance must be compressed into a fixed-size node embedding vector. A key nuanced point needed to understand information propagation across the sequence in Transformers is the fact they operate over a *causal* graph (or Markov chain), as mentioned before.

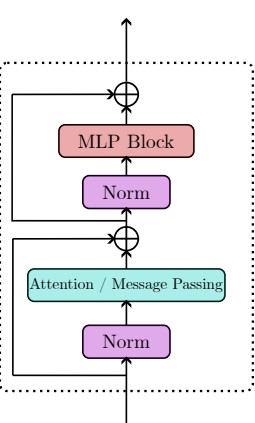

One of the main consequences emerging from the causal structure of the attention matrix is a tendency of earlier tokens to overwhelm later ones, a problem which was formally introduced in Barbero et al. (2024a). This can be interpreted as a form of the over-squashing problem, where in the limit in which the context size grows, only the first token will have an impact on the output of the model. Beyond this path-counting effect, Barbero et al. (2024a) also prove the appearance of *representational collapse*, where for some distinct sequences (e.g., a string that ends with a certain token versus the same string with that token repeated at the end), the last-token representations become arbitrarily close as length grows, rendering the model effectively unable to distinguish them under finite precision. This collapse is exacerbated by quantization/low-precision arithmetic and helps explain systematic failures on copying and counting in long, repetitive contexts. We present the formalization of these below.

Figure 5: Illustration of "Transformer-style" blocks in both Transformers and GNNs, where either Attention or message-passing acts as a "mixing" step.

**Theorem 4.2.4** (Representational Collapse (Barbero et al., 2024a))**.** *Let $\mathbf{v}^{(0)} \in \mathbb{R}^{n \times d}$ be a sequence and $\mathbf{v}^{*(0)} \in \mathbb{R}^{(n+1) \times d}$ be another sequence equal to $\mathbf{v}^{(0)}$ with the last token of $\mathbf{v}^{(0)}$ repeated. Assume that the positional encoding information decays to $0$ with the distance. Then, their representations become arbitrarily close as $n$ increases.*

**Theorem 4.2.5** (Over-squashing in Transformers (Barbero et al., 2024a))**.** *Consider an input sequence $\mathbf{v}_1^{(0)}, \ldots, \mathbf{v}_n^{(0)}$. Let $C > 0$ be some constant and $\bar{\alpha}_{i,j}^{(\ell)} = \frac{\alpha_{i,j}^{(\ell)}}{\beta_2} + \delta_{i,j}$, then:*

$$\left\| \frac{\partial \mathbf{y}_n}{\partial \mathbf{v}_i^{(0)}} \right\| \leq C \sum_{k_1 \geq i} \cdots \sum_{k_L \geq k_{L-1}} \bar{\alpha}_{n,k_L}^{(L-1)} \prod_{\ell=2}^{L-1} \bar{\alpha}_{k_\ell, k_{\ell-1}}^{(\ell-1)} \bar{\alpha}_{k_1, i}^{(0)} \tag{30}$$

Besides the issue of representational collapse, the fact that each token must integrate information from all previous tokens in the sequence leads to additional problems. In particular, we highlight the phenomenon of *over-mixing* outlined in Barbero et al. (2025), where the model thus loses the ability to distinguish

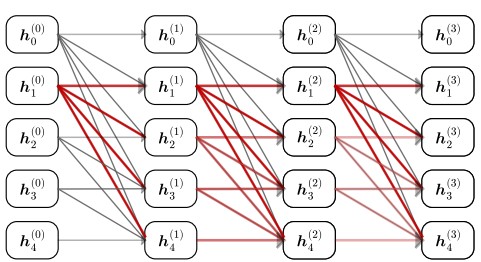 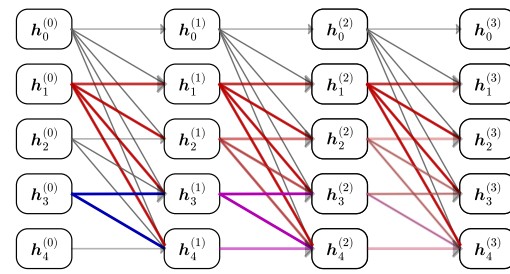

Figure 6: Illustration of information over-squashing in Transformers, adapted from similar figures in Barbero et al. (2024a; 2025). **Left:** Information in a decoder-Transformer "overmixes", which results in issues such as high sensitivity to perturbations early in the sequence. **Right:** The causal graph topology associated with a decoder-Transformer causes earlier tokens in the sequence (whose impact on later representations are depicted in red) to representationally "overwhelm" later ones (whose impact on later representations are depicted in blue)

.

individual tokens in the sequence and becomes more exposed to input perturbations. In the next subsection, we will present a number of attention patterns which large-scale models naturally learn through the gradient descent process to prevent this phenomenon. Illustrations of information over-squashing and over-mixing can be found in Figure 6, where the impact of a given token on later tokens is visualized by color propagating throughout the model.

Finally, we highlight that complementing these characterizations, a growing literature documents additional long-context failure modes and practical mitigations in decoder-Transformers in relation to attention sinks. For instance, training-free attention interventions can reduce extreme-token dominance and improve long-context stability (Han et al., 2025), while geometric and optimization viewpoints analyze how sink-like behavior emerges in attention dynamics (Ruscio et al., 2025; Alcalde et al., 2025). Related work also connects attention-sink dynamics to head collapse and proposes sink-aware training mechanisms to restore stable long-context routing (Fu et al., 2026), and other architectural proposals explicitly modulate the attention branch to mitigate extreme-token phenomena (Bu et al., 2025; Fu et al., 2026).

### 4.3 Shared Mitigation Strategies: Preventing Collapse in Depth and Width

Previously, we have introduced the over-smoothing and over-squashing issues, and how they manifest in both decoder-Transformers and GNNs. In this subsection, we will emphasize how and why different architectural decisions can prevent or mitigate these phenomena, and what shortcuts models naturally learn during training to protect themselves against these issues.

**Preventing Collapse in Depth: The Importance of the Transformer Block.** As we have discussed, naively stacking message-passing or self-attention layers typically leads to trainability issues and problems with diversity of representations, even after a few layers. In the case of Transformers, self-attention layers are typically not stacked one after the other, but interleaved with normalization and residual layers to form a *Transformer block*, which is depicted in Figure 5.[6] On the other hand, GNN layers were originally naively stacked one after the other without placing a similar emphasis on architectural optimization. While this choice was likely due to the locality present in some of the early GNN benchmarks, such as citation networks (McCallum et al., 2000; Sen et al., 2008), it could have been one of the culprits of the well-reported issue of over-smoothing in the GNN community.

---

[6]This Figure is intended as a general schematic of Transformer-style stabilization block, rather than a claim that there is a single canonical block used across architectures. For Transformers, canonical variants include post-LN (Vaswani et al., 2017) and the now-standard pre-LN formulation (Xiong et al., 2020). In GNNs, deep architectures similarly adopt residual connections and normalization, but implementation details can differ materially; in particular, dropout and normalization are often applied after message passing (e.g., in the implementations of Luo et al., 2024).

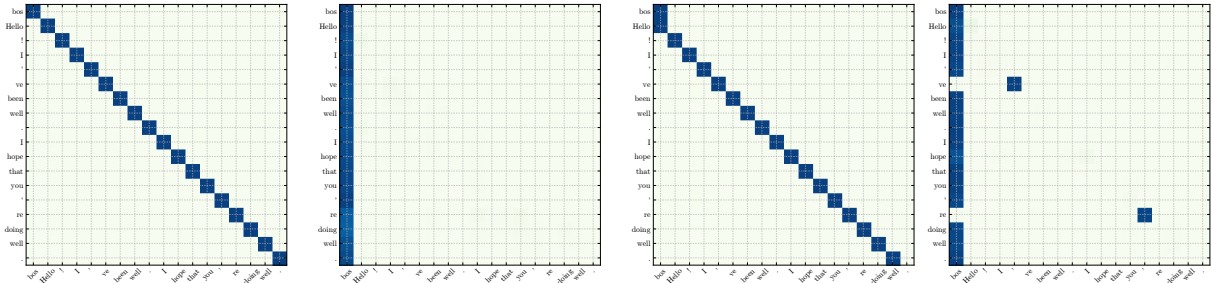

Figure 7: Example of sharp attention matrices in the Gemma 7B model. **Left:** An identity head. **Middle Left:** A BoS head. **Middle Right:** A previous token head. **Right:** An apostrophe head.

We highlight that the specific design of the Transformer block is critical for mitigating the undesirable contraction properties that can characterize the self-attention matrix, and facilitates signal propagation through the network, and hence the overall training process. In the original Transformer architecture (Vaswani et al., 2017), the LN block was placed after the residual connection (also known as post-LN). However, in this original post-LN setup, large gradients at initialization force the use of a warm-up phase to prevent instability. As such, Xiong et al. (2020) proposed to reposition the LN inside the residual branch, (pre-LN), which led to well-behaved gradients that neither explode nor vanish even with a large initial learning rate, thus removing the necessity for a warm-up stage.

Despite the success of the block structure in Transformers, we highlight that it is also possible to achieve good signal propagation without relying on this specific structure. An example of this is He et al. (2023), where the use of bias matrices, location-dependent rescaling, and kernel-preserving initializations that let *vanilla* Transformers (*without* skip connections or normalization) maintain unit-variance signal all the way through hundreds of layers, enabling effective signal propagation. Their follow-up study (He & Hofmann, 2023) goes further by pruning value projections, merging the attention and MLP sequence into a single operation, and discarding LayerNorm, and they obtain blocks that are 15% faster and 15% smaller yet match the per-update training speed and final accuracy of standard pre-LN stacks. We also highlight the recent work of Zhu et al. (2025), who replace layer normalization with the *Dynamic Tanh (DyT)* activation function in the transformer block while achieving comparable performance.

The extent to which individual elements of the Transformer block prevent rank collapse has been studied in a growing theoretical literature, including in some of the papers already mentioned. At the level of the mixing operator itself, Dong et al. (2021) show that attention-only stacks exhibit rapid (doubly-exponential) rank decay with depth, motivating the need for architectural "stabilizers" beyond pure attention. From a training and signal-propagation perspective, Noci et al. (2022) argue that appropriate depth-dependent scaling of the residual branches can restore stable propagation, a perspective recently echoed in Joseph et al. (2024). Complementarily, Wu et al. (2024) analyze LayerNorm and attention masking change the collapse dynamic, and show that although masking alone does not remove collapse, sparsity/locality in masks can slow it down, and that LayerNorm can materially alter the dynamical system, admitting richer non-collapsed behaviors for suitable parameter regimes. On the other hand, Naderi et al. (2024) attribute collapse and propagation pathologies to outlier spectral structure of attention operators, and propose simple mitigation principles that effectively temper these dominating modes. Finally, recent work further sharpens what it means to "fix" collapse via the block: Alman & Song (2025) argue that skip connections alone are not sufficient under small-weight regimes, introducing a related notion of layer collapse and concluding that sufficiently large weights are necessary for expressive depth even in the presence of residual pathways.

Analogously, the GNN literature has converged on a canonical "deep GNN block" that mirrors the Transformer block's intent of preserving stable gradient propagation across many mixing steps while preventing feature collapse. A number of modern GNNs (Li et al., 2019; 2020) typically wrap message passing inside residual/skip pathways, and augment it with normalization and regularization that explicitly control the contraction induced by repeated adjacency mixing (Rong et al., 2019). Concretely, methods like PairNorm (Zhao & Akoglu, 2019) (and related feature-normalization schemes) directly counteract over-smoothing by

maintaining representation variance across nodes. Importantly, many of these mechanisms instantiate the same two primitives that are central to the Transformer block, an identity (skip) pathway plus normalization, suggesting a shared "trainability template" for deep mixing architectures, see also (Scholkemper et al., 2024). Interestingly, the application of similar design principles in GNNs has yielded impressive, even state-of-the-art, performance (Luo et al., 2024; 2025). We note, however, that the specific instantiation of these blocks in GNNs is not identical to the standard Transformer block: in particular, GNN implementations often place normalization and dropout at slightly different locations. Nevertheless, the underlying design principle (combining identity pathways with normalization to stabilize repeated mixing) remains closely aligned across the two settings. See Figure 8 for an illustrative comparison of how adding residual connections and normalization/scaling reshapes the layer-wise Jacobian eigenspectrum, shifting it toward the identity and suppressing unstable modes. [7]

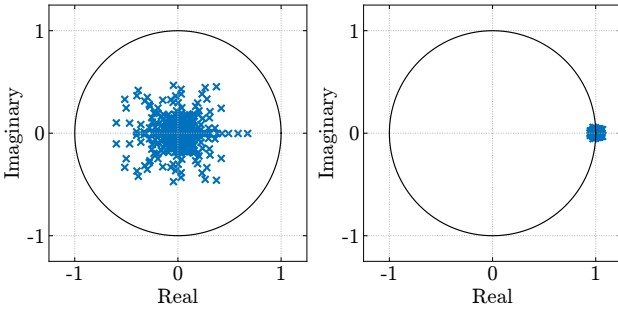

Figure 8: Illustration of the stabilizing effect of residual connections and normalization on GNN dynamics. **Left:** Eigenspectrum of the layer-wise Jacobian of a vanilla GCN, where repeated application of the normalized adjacency induces strong contraction and concentrates eigenvalues near the origin, leading to vanishing gradients and feature collapse at depth. **Right:** Eigenspectrum of a GCN augmented with a residual connection and output scaling. The residual pathway introduces an identity component that shifts eigenvalues toward $(1, 0)$, while normalization/scaling controls spectral spread and suppresses unstable (exploding) modes. Together, these mechanisms stabilize signal and gradient propagation and mitigate depth-induced collapse. Adapted form Arroyo et al. (2025).

At the same time, it is worth noting that, until very recently, the GNN community often framed depth limitations primarily through a representational lens (over-smoothing), with optimization and gradient-flow considerations receiving comparatively less emphasis. Arroyo et al. (2025) revisited this idea by explicitly linking over-smoothing/zero-collapse to vanishing gradients and trainability, thereby placing GNN block design on the same mechanistic level as signal-propagation analyses in deep Transformers. Overall, we believe this alignment shows that Transformer-style block design is not unique to attention; rather, it is a general recipe for making mixing and pointwise update models trainable at depth while limiting collapse.

Other research has identified methods established in the GNN and Graph Signal Processing literature that can be adapted to mitigate depth collapse in Transformers. Choi et al. (2023) introduce a modified attention mechanism inspired by graph filters, to encourage retention of higher frequencies and avoid over-smoothing. Kim & Ko (2024) similarly introduce graph propagation concepts in order to more faithfully capture local dependencies in vision Transformers.

**Preventing collapse in width: Sharpness and gating help with over-squashing.** One of the most effective strategies Transformers can learn to prevent this representational bottleneck is by developing specialized, *sharp* attention heads. Instead of forming a densely connected graph, these heads learn sparse patterns that route information in a more structured way, effectively preventing over-mixing. To this end, language models empirically tend to construct attention heads that either attend to very specific patterns in the sequence (previous token, spaces, or apostrophes), or attend only to the *BoS* (Beginning of Sequence) token despite not having any semantic meaning. This can be clearly seen in the example heads from the Gemma 7B model shown in Figure 7, which have also been reported empirically in Barbero et al. (2025); Queipo-de Llano et al. (2025). This "attention sink" phenomenon was originally reported in Xiao et al. (2023), and analyzed in more detail in Gu et al. (2024); Barbero et al. (2025). We note that recent work

---

[7]We also note that Graph Transformers are commonly instantiated using the same block-level stabilization template as sequence Transformers while injecting graph inductive bias via structural/positional encodings and, in hybrid variants, additional local message-passing modules (Ying et al., 2021; Rampášek et al., 2022; Ma et al., 2023).

(Qiu et al., 2025) also prevented the appearance of attention sink by incorporating *gating* into the attention mechanism, which can be seen as an alternative (and more direct) way of preventing over-mixing. While these ideas have been heavily explored in the context of recurrent architectures (Hochreiter & Schmidhuber, 1997; Gu & Dao, 2023; Beck et al., 2024; De et al., 2024), it appears that similar approaches can mitigate the undesirable effects of over-mixing in decoder-Transformers. We highlight also other works that identify other "width" issues arising from long contexts. Barbero et al. (2024b) examine RoPE and discover that it introduces issues when generalizing to long contexts. Xiong et al. (2025) examine attention "biases" in long contexts and design a KV caching technique to reduce the effect of these biases. Further, related empirical studies also document systematic degradation in long-context use, such as retrieval failures where evidence placed in the middle of the prompt is underutilized (Liu et al., 2024).

A useful way to make this precise is to view influence propagation through a decoder-Transformer as a sum over causal paths in the attention graph. The following bound (Barbero et al., 2025) makes explicit how attention weights determine the sensitivity of a late token to an early token through products of mixing coefficients:

**Theorem 4.3.1** (More detailed over-squashing bounds. (Barbero et al., 2025)). *Let $C_{max} > 0$ be the greatest Lipschitz constant of any layer of the Transformer, $H$ be the number of heads, and $\delta_i^j$ be 1 iff $i = j$ and 0 otherwise. Let $k \in \mathcal{P}_{i \to j}$ be a path from $i$ to $j$ of length $L$. Set $\bar{\alpha}_{ij}^{(\ell)} = \sum_h \alpha_{ij}^{(\ell,h)} + \frac{\delta_i^j}{H}$. Then:*

$$\left\| \partial \mathbf{v}_j^{(L)} / \partial \mathbf{v}_i^{(0)} \right\| \leq C_{max}^L \sum_{k \in \mathcal{P}_{i \to j}} \bar{\alpha}_{j,k_{L-1}}^{(L)} \bar{\alpha}_{k_{L-1},k_{L-2}}^{(L-1)} \dots \bar{\alpha}_{k_1,i}^{(1)}. \tag{31}$$

This bound exposes the mechanism behind width failures: the influence $\left\| \partial \mathbf{v}_j^{(L)} / \partial \mathbf{v}_i^{(0)} \right\|$ is controlled by a path-sum of products of attention weights. When attention is diffuse (many moderate $\alpha_{uv}$), the number of contributing causal paths can grow rapidly and the model can overmix: many tokens become entangled through many comparable routes, amplifying sensitivity to small perturbations and pushing representations toward a common subspace. Conversely, when attention is sharp, most paths have near-zero weight, so the path-sum is dominated by a small set of routes, which reduces mixing. Likewise, gating effectively downweights the attention branch in the path products, which reduces cumulative mixing as well.

The ability of decoder-Transformers to reliably learn and maintain these desirable sharp patterns is also fundamentally constrained by the choice of attention function. To this end, recent work (Veličković et al., 2024) argue that the standard `softmax` will fail in settings that require generalization to sequences longer than those seen during training. In these cases, the nature of softmax can cause attention scores to bleed and diffuse, making it difficult for a head to remain sharply focused on a single token. We highlight other works which have further analyzed the softmax function (Masarczyk et al., 2025) as well as other variants (Saratchandran et al., 2024; Ramapuram et al., 2024), including ones that prevent the formation of attention sinks (Zuhri et al., 2025). We note that Over-squashing in GNNs is also sometimes addressed through methods that reduce bottlenecks in the computational tree. One recent method to do this has been *adaptive message-passing* (Errica et al., 2023; Finkelshtein et al., 2024), which can also be seen as a more explicit way to enforce "sharpness".

## 5 Conclusion

In this survey, we have explored the conceptual and mathematical parallels between decoder-Transformers and GNNs, advocating for a unified graph-based perspective on information propagation in these seemingly disparate architectures. We have shown how fundamental challenges in GNNs, such as **over-smoothing** and **over-squashing**, find direct analogues in Transformers as **rank collapse** and **representational collapse**.

Our analysis revealed that the iterative message-passing operations in GNNs, which can lead to node feature homogenization, mirrors the depth-wise rank collapse observed in Transformers where token representations lose their distinctiveness. The role of residual connections and normalization techniques, crucial for mitigating these issues in both architectures, underscores a shared need for mechanisms that preserve signal

diversity and prevent information dissipation or concentration. Furthermore, the sensitivity bounds derived for GNNs provide a lens through which to understand how information from distant parts of a sequence can be "squashed" in Transformers, highlighting the importance of managing receptive field and information flow.

By viewing self-attention/message passing as the repeated application of a mixing–update operator, our perspective makes explicit a mechanistic link between propagation properties and architectural choices. In depth, when the operator is contractive, iterating it drives representations toward fixed points and suppresses variation, aligning with rank/feature collapse; residual pathways, normalization, and scaling act by injecting/maintaining an identity component and controlling contraction, improving trainability. In width, influence between distant tokens/nodes is governed by path-sums of products of mixing coefficients; sharp attention patterns and gating reduce the number/weight of active paths, mitigating over-mixing and sensitivity bottlenecks. Overall, the contribution is a unifying propagation lens that organizes existing failure modes and clarifies how common stabilizers intervene, while also suggesting concrete directions for more principled mitigation design.

Ultimately, this unified view not only enhances our theoretical understanding of these powerful models but also paves the way for cross-pollination of ideas. Solutions developed for performant GNNs, particularly those addressing vanishing gradients and information bottlenecks, could inspire novel architectural designs or training strategies for Transformers, and vice-versa. As the fields of graph representation learning and large language models continue to advance, a shared vocabulary and conceptual framework will be valuable for developing more robust, efficient, and interpretable deep learning architectures.

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
