# OpenReview forum: "A Survey on Over-smoothing and Over-squashing: Unified Propagation Perspectives on Graph Neural Networks and Transformers"
_TMLR — Accepted by TMLR_

### Review · Reviewer_onyQ · 2026-01-09

**Summary Of Contributions:**

This paper presents a survey connecting GNNs and decoder-Transformers through a unified graph perspective. The main contribution is showing how problems studied in GNNs (over-smoothing, over-squashing) map onto issues in Transformers (rank collapse, representational collapse). The authors argue that the self-attention mechanism can be viewed as a learned adjacency matrix, which means insights from one field can transfer to the other.

  Key strengths:
  - The unified framework is intuitive and well-motivated
  - Good coverage of both GNN and Transformer literature
  - The figures (especially Fig 1, 4, 7) help explain the concepts clearly
  - Practical discussion of mitigation strategies (residual connections, normalization, sharp attention)

  Weaknesses:
  - Mostly a literature review, limited new theoretical results
  - Some sections feel a bit dense with citations without enough synthesis
  - Missing discussion of computational trade-offs between different approaches

**Additional Comments:**

Overall this is a solid survey paper that provides a nice unified lens for understanding Transformers and GNNs. The writing is clear and the structure is logical. My main suggestion would be to go a bit deeper on the practical implications - what should practitioners actually do differently based on these insights? But I understand space constraints. The figures are well done and help a lot with understanding the concepts.

**Audience:**

Yes

**Audience Explanation:**

Yes, I think this will be useful for researchers working at the intersection of GNNs and LLMs. The unified perspective could help practitioners diagnose issues in their models and potentially transfer solutions across domains. Given the current interest in understanding Transformer internals and scaling behavior, a principled framework connecting them to better-understood GNN dynamics seems timely.

**Broader Impact Concerns:**

None. This is a theoretical/survey paper examining architectural properties of neural networks. I don't see any direct ethical concerns.

**Claims And Evidence:**

Yes

**Claims Explanation:**

The paper is primarily a survey, so the claims are well-supported by existing literature. The authors do a reasonable job citing the relevant work from both GNN and Transformer communities. The visualizations (attention matrices from Gemma 7B, eigenvalue histograms) provide empirical grounding for the theoretical arguments. I didn't spot any obvious mischaracterizations of prior work.

**Requested Changes:**

1. (Minor) Section 3 and 4 could benefit from a clearer distinction between what's established theory vs. recent observations. Some parts blend these together.
  2. (Minor) The paper would be stronger with a concrete example or case study showing how a technique from GNNs was successfully applied to improve a Transformer, or vice versa. Right now it's mostly "these things are similar" without demonstrating practical cross-pollination.
  3. (Minor) Table summarizing the key correspondences (over-smoothing ↔ rank collapse, over-squashing ↔ representational collapse, etc.) would help readers quickly grasp the main mapping.
  4. (Minor) Some newer relevant work on long-context LLMs and their attention patterns could be added to Section 4.2

---

> ### Author Response · Authors · 2026-02-07
> **Reply to the Reviewer (1)**
>
> We thank the reviewer for the thoughtful and encouraging assessment. We are glad the unified graph perspective and the cross-community mapping came across clearly, and we appreciate the concrete suggestions to improve synthesis and practical takeaways. In the revision, we have (i) sharpened the distinction between established theory vs. recent empirical observations in Sections 3–4, (ii) added a compact set of bullet points to signpost the main contributions, and (iii) expanded the discussion of cross-pollination and mitigations with additional recent references.
>
> > (Minor) Section 3 and 4 could benefit from a clearer distinction between what's established theory vs. recent observations. Some parts blend these together.
> >(Minor) Table summarizing the key correspondences (over-smoothing ↔ rank collapse, over-squashing ↔ representational collapse, etc.) would help readers quickly grasp the main mapping.
>
> These are both good points, and echo the comments of other reviewers about clarity of presentation. We have added a list of bullet points to summarise our contributions and signpost the key messages of the text.
>
> >(Minor) The paper would be stronger with a concrete example or case study showing how a technique from GNNs was successfully applied to improve a Transformer, or vice versa. Right now it's mostly "these things are similar" without demonstrating practical cross-pollination.
>
> This is a good suggestion: we are adding related work with models that inject graph inductive bias / message passing into Transformer blocks, including Choi et al. (2024) and Kim & Ko (2024)
>
> >(Minor) Some newer relevant work on long-context LLMs and their attention patterns could be added to Section 4.2
>
> We agree, and we have updated Section 4.2 to include newer work on long-context LLM attention patterns and mitigations, which can be found at the end of the section.
>
> ## References
>
> Kim & Ko 2024: Rethinking Attention Mechanisms in Vision Transformers with Graph Structures. https://www.mdpi.com/1424-8220/24/4/1111
>
> Choi et al. 2024: Graph Convolutions Enrich the Self-Attention in Transformers! https://arxiv.org/abs/2312.04234v5

---

### Review · Reviewer_JEi5 · 2026-01-17

**Summary Of Contributions:**

This work is motivated by the identification of common properties between message-passing Graph Neural Networks and decoder-Transformer architectures.
The manuscript is organized into three parts:
- **Section 2** provides a foundational overview of sequence modeling (using RNNs and Transformers) and message-passing GNNs;
- **Section 3** describes in-depth the concepts of over-smoothing, vanishing gradients and over-squashing, overviewing the existing literature within GNN field;
- **Section 4** proposes a unified representation of Transformers and MPGNNs. It specifically addresses how Transformers mitigate rank and representation collapse.

## Strengths

1. Section 3 provides an in-depth overview of over-smoothing, vanishing gradients, and over-squashing within the GNN domain.
2. Section 4.3 lists current methods to prevent rank and representation collapse in the Transformer domain.


## Weaknesses
(see more details in the Requested Changes section)

1. The manuscript lacks valuable insights, which can be demonstrated by either:
    * Comprehensive literature review *OR*
    * Extensive experiments showing practical mitigation of over-smoothing or over-squashing concepts in transformers *OR*
    * Theory for presented concepts when applied to transformers
2. While the paper defines a connection between message passing GNNs and Transformers, this unified view is not leveraged to generate new insights or drive the subsequent discussion.
3. The paper fails to address the existing body of literature on Graph Transformers. This literature must be addressed when graph and transformer concepts are discussed together.
4. The authors dedicate more than one page out of thirteen to the mathematical foundations of RNNs in Section 2, yet these are never addressed in the remainder of the paper.
6. The motivation and discussion are overly focused on Large Language Models (LLMs). The proposed "unified view" is not specific to LLMs. Thus, the paper lacks a connection to vision transformers, time-series transformers, graph transformers, etc.

**Audience:**

No

**Audience Explanation:**

In its current state, the paper is unlikely to be of interest for the TMLR's audience.
It lacks either a comprehensive literature review, *OR* theoretical contributions, *OR* rigorous empirical results demonstrating how GNN-based insights can be substantively applied to Transformer architectures.

The manuscript is inadequate as a cohesive survey.
While Section 3 provides a comprehensive literature overview of over-smoothing, vanishing gradients, and over-squashing in GNNs, it lacks an analysis of how these issues are measured, the specific tasks where they occur, or a taxonomic approach to mitigation strategies.

The paper presents no theoretical analysis.
The purported connection between over-smoothing and rank collapse is not established in a technically sound manner.
Furthermore, the discussion regarding over-squashing and representational collapse is merely a brief summary of theoretical findings from Barbero et al. (2024), omitting details and leading to no new insights.

There are no empirical experiments to support the authors' claims.
While Section 4.1 outlines mathematical similarities between message passing GNNs and Transformers, this "unified view" does not result in a novel architectural design or a comparative demonstration of shared properties.
The manuscript fails to discuss how this unified view can be leveraged to improve Transformer.

**Broader Impact Concerns:**

No ethical concerns

**Claims And Evidence:**

No

**Claims Explanation:**

**NOTE:** The paper is not fully clear regarding its primary claims (see requested changes). Here, I evaluate claims extracted from the abstract, introduction, and conclusion.

1. &#x2705; "we present a unified graph perspective that bridges the theoretical understanding of decoder-Transformers and GNNs" and "interpreting the self-attention mechanism as a learned adjacency matrix, we bridge the gap between spectral graph theory and the dynamics of attention."
This claim is largely met. Equations (26)-(28) in Section 4 establish a unified mathematical view of GNNs and Transformers.
2. &#x274C; "We systematically examine how well-known phenomena in GNNs, such as over-smoothing and over-squashing, directly manifest as analogous issues like rank collapse and representational collapse in deep Transformer architectures."
The paper provides no systematic experimental evidence to support this claim. The lack of empirical validation is a weakness.
3. &#x2753; "we reveal shared underlying principles governing signal propagation"
The formulation of this claim is ambiguous. On the one hand, section 4.1 shows that a self-attention block can be viewed as a Graph Attention Transformer layer (GAT). At the same time, GAT is founded on the same self-attention math that is later used in self-attention block of the transformer. Veličković et al. (2018) explicitly define GAT as self-attention.
Therefore, the paper does not "reveal" a new principle, but rather restates an existing one.
4. &#x274C; "[we] demonstrate how insights from one field can illuminate challenges and solutions in the other" and "[we aim] to highlight areas for cross-pollination of research"
This claim is vaguely formulated and arguably is not met. For example, let us look at the "The Depth Regime" subsection of Section 4.2. While the text notes conceptual similarities between rank collapse and over-smoothing based on the eigenspectrum, it fails to demonstrate how GNN research provides actionable solutions for Transformer rank collapse. The Gemma 7B experiment in Section 4.2 illustrates the rank collapse problem but offers no cross-disciplinary solution.
5. &#x2753; "We analyze the role of architectural components like residual connections, normalization, and causal masking in these issues."
This claim is somewhat true. Although the paper discusses components that prevent these issues and cites relevant literature, it lacks empirical or theoretical analysis.
6. &#x274C; "We aim to provide a framework for understanding how information flows through deep learning models that perform sequence mixing through an adjacency operator"
The manuscript overviews existing works on information flow in message-passing GNNs, but it fails to synthesize these into a distinct, cohesive framework.
8. &#x2705; "we elaborate on how the normalized adjacency matrix – the key component of mixing architectures – affects information flow in depth (in the number of layers) and width (in the diameter of the graph or sequence)"
This is sufficiently discussed in Section 3.
9. &#x2705; "we have explored the conceptual and mathematical parallels between decoder-Transformers and GNNs, advocating for a unified graph-based perspective on information propagation in these seemingly disparate architectures."
While I disagree with the statement that "Transformers and GNNs [are]...seemingly disparate architectures" (see below), this claim is technically fully met in Section 4.1 and Section 4.2.
10. &#x274C; "Our analysis revealed that the iterative message-passing operations in GNNs, which can lead to node feature homogenization, mirrors the depth-wise rank collapse observed in Transformers where token representations"
The analysis in Section 4.2 is superficial.
I must highlight that a convincing link between concepts requires more than semantic similarity (e.g., both having "contractive" components).
It requires a rigorous demonstration of empirical or theoretical similarities.
The fact that GNNs have "the *contractive nature* of the mixing matrix's spectrum" and for Transformers "many learned attention heads are, in fact, highly *contractive*" does not fit such a demonstration of similarity.
11. &#x274C; "this unified view ... enhances our theoretical understanding of [transformers]"
While a unified view has the potential to enhance theoretical understanding of transformers, this manuscript presents no new theoretical results to achieve this goal.

**Requested Changes:**

**IMPORTANT:** I see three ways how this work can be strengthened: survey route, theory route, and experimentation route.
To secure my positive recommendation, the authors should address my comments from a "common" critical section and *ONE* route of their choice (Survey, Theory, or Experimentation).
This will provide the depth currently missing from the manuscript.

# Common Critical Changes (Required)

The paper relies on the premise that there are "structural similarity between GNNs and Transformers." However, **Transformers are GNNs**. The paper completely ignores the field of Graph Transformers, which studies how to process a graph with a transformer. To demonstrate how extensive this field is, I encourage authors to check out a survey on Graph Transformer by Shehzad, et al. (2024). Authors must incorporate some literature on Graph Transformers, at the very least mentioning the known ones, like Rampášek et al. (2022), Ma et al. (2023), and Wu et al. (2022).

Why is the paper so focused on LLMs? It seems arbitrary. Not a single concept in Section 2-4 is exclusive to LLMs. Yes, you may mention that LLMs had "the remarkable success" in recent years. But you also need to mention that there are the other types of transformers with different specificity (Vision, Time-Series, and Graph. For other types, you may consult the survey by Islam et al. (2024). The authors must either explicitly justify the exclusive focus on LLMs *OR* make the motivation more general.

The current claims are formulated vaguely and are scattered over the text. Perhaps, the authors can summarize the contributions in a list *OR* write an explicit problem statement section. This will make it easier for a reader and increase the quality of your work.
For example, "In this paper, we present a unified graph perspective that **bridges the theoretical understanding** of decoder-Transformers and GNNs." (from the Abstract).  Which part of the paper presents said bridging theory? Such a formulation makes the reader expect a specific theoretical result.

Sections 4.1–4.2 identify similarities between problems in GNN domain and problems in Transformer domain. Can authors discuss in depth how analysis from GNN field listed in Section 3 can be used to improve transformer's performance, training stability, etc.?
For example, the most direct application of GNNs can be by taking a Transformer block (say, in Gemma 7B) and replacing MHA with the GNN layer and a causal graph. As a matter of fact, Choi et al. (2024) and Kim & Byoung (2024) present updates to a general Transformer structure with exactly this approach.

Section 4.3 is called "Shared Mitigation Strategies" and promises to introduce how "they manifest in both decoder-Transformers and GNN". Despite its title, this section focuses almost entirely on collapse-preventing strategies for Transformers. The author must include the discussion of how the similar mitigation strategies manifest in GNNs. The authors should discuss what are the similarities and differences of these strategies.

Why is RNN background literature covered? In this work, no similarities between RNN and GNN are leveraged. The inclusion of RNN mathematical foundations is currently a distraction for the reader. The RNNs are only briefly mentioned in Section 4.3 in the context of gated attention, which does not require such an extensive background. The authors have to either integrate RNNs into the analysis *OR*  remove this subsection.

Page 10. The histogram of its associated eigenvalue modulus shown in Figure 4 is a trivial consequence of the attention matrix being row-stochastic. The row-stochastic matrix always has eigenvalue modulus upper-bounded by one. This fact is directly acknowledged on Page 11 of the text. Why is this fully expected result significant and relevant? Could you provide a more nuanced analysis of this figure?

Page 12. While the block on Figure 6 may be a valid block, it is important to cite the paper this block is taken from (both Transformer literature and GNN literature). There is a huge variation in the design of the transformer block.
For example, your block is different from the Transformer block by Vaswani et al. (2017).
This block design is also not very typical for message-passing GNNs, which rely on Dropouts and layerwise attention. Beside that, dropout and normalization in GNNs are commonly applied after the message passing (as an example, see the GNN implementation in Luo et al. (2024)).


# Critical Routes (Choose One)

## Survey route
**ROUTE NOTE:** Currently, some sections focus either exclusively on GNNs or exclusively on Transformers. The main idea is to survey the literature to further establish similarities between them.

Systematize the prevention methods for over-smoothing (GNN) vs. rank collapse (Transformer) and over-squashing (GNN) vs. representational collapse (Transformer). Section 3 contains a great overview of over-smoothing, vanishing gradients, and over-squashing concepts in GNNs. It lists a number of works that are analysing these concepts. Can you summarize what are the nature of the main methods for preventing them? Can you **survey Transformer papers** that are analysing and preventing the respective concepts?

For example, let us consider over-smoothing (GNN) vs. rank collapse (Transformer).
What are the top methods for preventing over-smoothing?
What are the top methods for preventing rank collapse?
Do such methods have anything in common?
What are the differences?
If any, why and what can be borrowed?
The text answering these questions will constitute a survey that bridges GNN and Transformers.
Make the same analysis for over-squashing in GNN and representation collapse in Transformers

Currently, your survey on over-squashing (GNN) vs. representational collapse (Transformer) consists on only two papers. Are there no works analysing over-squashing in GNNs? Barbero et al. (2024) and Barbero et al. (2025) have more than 50 citations in total. Does none of those works propose an additional analysis for representational collapse or over-mixing?

Section 4.3 discusses the design of a Transformer block. You need to add a discussion of a GNN block design. How GNN block is structured? What are the similarities in design between these blocks? Are the designs guided by the similar issues? Do the similarities in design result in the similar properties? Can the transformer block be enhanced with methods from Luo et al. (2024) (similarly to a classic GNN)?

In addition to a regular Transformer, you have to discuss the design of a graph transformer block, since it is bridging both fields. What are the design choices of Graph Transformer that are different from other Transformer blocks? Are there similarities between Graph transformer blocks and GNN blocks? How over-smoothing, rank collapse, over-squashing, and representational collapse manifest in Graph Transformer literature? What methods from GNN field are adapted by Graph Transformer to mitigate it?

## Theory route
**ROUTE NOTE:** Sections 4.1-4.2 vaguely point at similar mathematical properties for GNN and transformer, as well as referring to the paper analysing such properties from the theoretical point of view. Currently, this work does not introduce any theory. That makes analysis seem superficial.

Formulate a technically sound connection between over-smoothing and rank collapse. For example, could you show a sensitivity bound for Transformer similar to Theorem 3.2.1? Given the unified view from Section 4.1 and the theoretical connections and GNN literature from Section 3, can you claim something about the transformer? Can you show it with any toy experiment?

Look at how Barbero et al. (2024) formulates the theory connecting over-squashing and representational collapse. It is presented in two theorems with proofs. Can you make a similar connection for over-smoothing and rank collapse?

The theoretical connection for over-squashing and representational collapse can be cited directly from Barbero et al. (2024) appendix. However, it has to be incorporated into the presented theory.

The authors must discuss how the specificity of the graph domain and LLM structure affect the presented analysis. One of the main problems with applying spectral graph theory to LLMs is that spectral graph theory is mainly done for **undirected** graphs. The attention matrix of LLM is making a **directed** graph due to causal masking. Therefore, the assumptions of the literature you cite might not hold. You have to be especially careful about assumptions of eigenvalues.

Make sure any lemmas, theorems, or propositions you add have rigorous mathematical proofs.

## Experimentation route
**ROUTE NOTE:** One way to show a connection between GNN and GNN interpretation of a Transformer is to experimentally demonstrate that similarity between behaviour of GNNs and Transformers.
Unlike for the theory route above, one does not require bringing theory for this route, yet the experiments have to be convincing.
At the moment, this paper does not feature experiments that illustrate similar properties of the unified view of Transformer and GNNs. Figures 4 and 7, which can be considered example results, are only showing the specific properties of Transformer.

Demonstrate side-by-side over-squashing in both Transformer and GNN. Discuss how graph transformers mitigate it.

Demonstrate side-by-side sharp attention in Transformer and adaptive message passing GNN (or GAT).

Demonstrate side-by-side over-smoothing and rank collapse.

Demonstrate that a Transformer block modified with GNN-inspired layers (per Choi et al., 2024) exhibits predictable behaviour.

Make sure you are using two-three GNNs and two-three different transformers in every experiment. Specifically, include GAT. It is based on the same design idea as the transformer and thus can make a relevant point of connection.

# Minor
Page 7. Figure 1 is a little misleading. First, it shows a tree, which is a very special case of graphs. I recommend presenting an arbitrary graph colouring a root node and 1-hop, 2-hop,... neighbourhoods (see Figure 1 in Zhang, et al. (2022) for the reference). Second, message passing GNNs and graph transformers propagate signal on a graph differently. Figure 1 is suitable for message-passing GNNs. I recommend pairing it side-by-side with a signal propagation picture for the transformer.

Page 8. The colour scheme on Figure 3 is hard to distinguish, especially when histograms are overlapping each other. Consider more contrastive colours.

Page 9. "Transformers and GNNs have traditionally been viewed as distinct architectures" is generally wrong, because graph transformers are a thing and have to be mentioned in this paper.

Page 11. "In turn, modern LLMs ... impose a triangular (causal) mask for next-token generation. This, together with the softmax function, makes it row-stochastic." Consider rephrasing this bit for better readability. First, makes what stochastic? Second, if we assume "it" is "attention matrix", then softmax is solely responsible for row-stochasticity. The presence of causal masking has nothing to do with it. The attention matrix is row-stochastic even without the mask.

Page 11. Causal transformers (like modern LLMs) are using mask to ensure the information flows only from the current input to future hidden states. Despite the masking is being mentioned on the same page, there is a factual error, where a causal graph is implied to be a fully connected graph.

Page 12. Verify that Xiong et al. (2021) is the citation for pre-LN transformers.
This paper does not seem to be about transformers.

# Literature
1. [Vaswani, Ashish, et al. "Attention is all you need." Advances in Neural Information Processing Systems (2017).](https://proceedings.neurips.cc/paper/2017/file/3f5ee243547dee91fbd053c1c4a845aa-Paper.pdf)
2. [Barbero, Federico, et al. "Transformers need glasses! information over-squashing in language tasks." Advances in Neural Information Processing Systems (2024).](https://proceedings.neurips.cc/paper_files/paper/2024/file/b1d35561c4a4a0e0b6012b2af531e149-Paper-Conference.pdf)
3. [Barbero, Federico, et al. "Why do LLMs attend to the first token?." Conference on Language Modeling (2025).](https://arxiv.org/pdf/2504.02732)
4. [Islam, Saidul, et al. "A comprehensive survey on applications of transformers for deep learning tasks." Expert Systems with Applications (2024).](https://arxiv.org/pdf/2306.07303)
5. [Rampášek, Ladislav, et al. "Recipe for a general, powerful, scalable graph transformer." Advances in Neural Information Processing Systems (2022)](https://proceedings.neurips.cc/paper_files/paper/2022/file/5d4834a159f1547b267a05a4e2b7cf5e-Paper-Conference.pdf)
6. [Choi, Jeongwhan, et al. "Graph convolutions enrich the self-attention in transformers!." Advances in Neural Information Processing Systems (2024).](https://arxiv.org/pdf/2312.04234v3)
7. [Kim, Hyeongjin, and Byoung Chul Ko. "Rethinking attention mechanisms in vision transformers with graph structures." Sensors (2024).](https://www.mdpi.com/1424-8220/24/4/1111)
8. [Veličković, Petar, et al. "Graph attention networks." International Conference on Learning Representations (2018).](https://openreview.net/forum?id=rJXMpikCZ)
9. [Zhang, Shichang, et al. "Graph-less neural networks: Teaching old MLPs new tricks via distillation." International Conference on Learning Representations (2022).](https://openreview.net/forum?id=4p6_5HBWPCw)
10. [Luo, Yuankai, Lei Shi, and Xiao-Ming Wu. "Classic gnns are strong baselines: Reassessing gnns for node classification." Advances in Neural Information Processing Systems (2024).](https://proceedings.neurips.cc/paper_files/paper/2024/file/b10ed15ff1aa864f1be3a75f1ffc021b-Paper-Datasets_and_Benchmarks_Track.pdf)
11. [Shehzad, Ahsan, et al. "Graph transformers: A survey." arXiv preprint arXiv:2407.09777 (2024).](https://arxiv.org/pdf/2407.09777)
12. [Ma, Liheng, et al. "Graph Inductive Biases in Transformers without Message Passing." International Conference on Machine Learning (2023).](https://proceedings.mlr.press/v202/ma23c/ma23c.pdf)
13. [Wu, Qitian et al. "NodeFormer: A Scalable Graph Structure Learning Transformer for Node Classification" Advances in Neural Information Processing Systems (2022).](https://proceedings.neurips.cc/paper_files/paper/2022/file/af790b7ae573771689438bbcfc5933fe-Paper-Conference.pdf)

---

> ### Author Response · Authors · 2026-02-07
> **Reply to the Reviewer (1)**
>
> We thank the reviewer for the exceptionally detailed and constructive feedback. The comments identify a substantial number of important issues (ranging from missing related literature and positioning, to clarifications of claims, figures, and citations) that will significantly strengthen the manuscript. We apologize that our response took some time, as we were working towards another deadline. We have incorporated the requested changes into the manuscript, and we hope the revisions address the reviewer’s concerns to their satisfaction.
>
> ## Common Critical Changes (Required)
>
> >The paper completely ignores the field of Graph Transformers, which studies how to process a graph with a transformer.
>
> This is a good point that our current draft omits any discussion of Graph Transformers, which we agree is relevant to the discussion. In the revision we have (i) added a Graph Transformers subsection to position our unified perspective relative to this body of work, and (ii) incorporated key representative papers and surveys (e.g., Shehzad et al., 2024; Rampášek et al., 2022; Ma et al., 2023; Ying et al. 2021).
> We have also revised the manuscript to explicitly acknowledge that (graph) Transformers can be viewed as a family of GNNs, and that our goal is to use this shared message-passing/mixing viewpoint to create a common language for depth/width pathologies and mitigation strategies across settings.
>
> > Why is the paper so focused on LLMs? It seems arbitrary. Not a single concept in Section 2-4 is exclusive to LLMs. … you also need to mention that there are the other types of transformers with different specificity (Vision, Time-Series, and Graph.
>
> We accept that our motivation text currently emphasizes LLMs, but we would like to clarify that not all of the concepts as discussed in sections 2-4 generalise to the other modalities you mention (especially vision and graph); some of the discussed phenomena are relevant – either primarily or exclusively – to causal Transformers, of which the most widely studied are LLMs. These phenomena include the outsized impact of earlier tokens due to over-squashing as observed by Barbero et al. 2024.
>
> However, it is true that many of the underlying propagation issues and architectural themes apply broadly to Transformers in vision, time-series, audio, and graphs. In the revision, we have (i) explicitly stated that our framework targets decoder-style causal Transformers as a convenient and widely-used instantiation where directed/causal masking makes propagation effects particularly relevant, while (ii) clarifying that the same “mixing via an adjacency operator” lens extends to other Transformer families and applications (e.g., ViTs, time-series Transformers, Graph Transformers). We have also added a short paragraph and citations to broad Transformer application surveys.
>
> >The current claims are formulated vaguely and are scattered over the text.
>
> We agree and have revised the manuscript to make the primary contributions explicit and easier to locate. Concretely, we have added a “Contributions and scope” bullet list in the Introduction that summarizes our main messages and points the reader to the formal development in Section 4.1.
>
> >Sections 4.1–4.2 identify similarities between problems in GNN domain and problems in Transformer domain. Can authors discuss in depth how analysis from GNN field listed in Section 3 can be used to improve transformer's performance, training stability, etc.?
>
> Thank you for bringing this to our attention. We would like to highlight that our intent in drawing the “attention-as-learned-graph-operator” equivalence was precisely to encourage this cross-pollination, as we find that this is a perspective that has not been widely adopted in the literature. In particular, once attention is treated as a (directed, data-dependent) graph operator, tools from graph signal processing and network science (e.g., contraction/Dirichlet-energy viewpoints, bottleneck measures and curvature-inspired notions, and rewiring principles) become natural candidates for (i) diagnosing training instabilities/vanishing gradients in depth and (ii) improving long-range communication in width, which we believe is a perspective that is not prevalent in the literature.
>
> At the same time, we acknowledge that our current draft was missing some relevant references, and we appreciate you highlighting these. We have expanded related work to include the models you highlight, which do already inject graph inductive bias / message passing into Transformer blocks, including Choi et al. (2024) and Kim & Ko (2024). In addition, we have cited and discussed recent work analyzing long-context attention pathologies. But we emphasise that this remains an under-explored field, and this gap is exactly what our survey aims to highlight and stimulate.

---

> > ### Author Response · Authors · 2026-02-07
> > **Reply to the Reviewer (2)**
> >
> > > Section 4.3 is called "Shared Mitigation Strategies" … The author must include the discussion of how the similar mitigation strategies manifest in GNNs. The authors should discuss what are the similarities and differences of these strategies.
> >
> > Thank you for bringing this to our attention. We have added a paragraph describing the canonical “deep GNN block” design and how it mirrors the Transformer block’s intent: maintaining stable propagation and preventing collapse via residual/skip pathways and normalization. We have cited deep-GNN training recipes (e.g., DeeperGCN) and normalization/regularization mechanisms (PairNorm etc.), and have discussed similarities/differences in where normalization and dropout are applied in typical GNN implementations.
> >
> > >Why is RNN background literature covered? In this work, no similarities between RNN and GNN are leveraged. …The authors have to either integrate RNNs into the analysis OR remove this subsection.
> >
> > The inclusion of RNN background is deliberate: classical RNN theory provides the basis (and historical motivation) for the discussion of Jacobian-product–driven phenomena such as vanishing/exploding gradients and edge-of-chaos behavior. This mechanism underpins much of our “shared pathologies” discussion throughout our work, and we suggest, therefore, that it requires a complete introduction. As such, we do intend to suggest a structural similarity between RNNs, GNNs, and attention-based models, following recent work that made this connection explicit via the vanishing gradient link (Arroyo et al. (2025)), which interprets residual connections as latent state evolution and analyzes the resulting Jacobian dynamics.
> >
> > Our contribution builds on this insight by extending the same lens to attention-based mixing operators, where the residual stream again defines an implicit "state". In this sense, the RNN background provides the conceptual starting point of a unifying structural view across recurrent, graph-based, and attention-based architectures.
> >
> > We agree, however, that in the current draft the role of the RNN background is unclear. In the revision, we have (i) highlighted the role of Arroyo et al. (2025) more clearly as the core work establishing this structural similarity, and (ii) added an explicit bridge paragraph explaining how the same Jacobian-based arguments apply across RNNs, GNNs, and Transformers.
> >
> > >Page 10. The histogram of its associated eigenvalue modulus shown in Figure 4 is a trivial consequence of the attention matrix being row-stochastic. The row-stochastic matrix always has eigenvalue modulus upper-bounded by one. This fact is directly acknowledged on Page 11 of the text. Why is this fully expected result significant and relevant? Could you provide a more nuanced analysis of this figure?
> >
> > We agree that the upper bound $|\lambda|\le$ 1 follows from row-stochasticity and is not itself the point. Our intent was to highlight an empirical fact: many learned heads are not merely bounded, but highly contractive (mass concentrated well below 1), which has direct implications for gradient flow and depth scalability, and helps motivate why careful block design (residual + normalization placement) is necessary in practice. We have revised the caption to make this explicit.
> >
> > > Page 12. While the block on Figure 6 may be a valid block, it is important to cite the paper this block is taken from (both Transformer literature and GNN literature) .... Beside that, dropout and normalization in GNNs are commonly applied after the message passing (as an example, see the GNN implementation in Luo et al. (2024)).
> >
> > Agreed. Figure 6 is intended as a schematic for “Transformer-style stabilization block” (Norm + Residual + Mixing/MLP), not as a claim that there is a single canonical block. We have (i) cited canonical block variants (post-LN vs pre-LN and modern practice)and (ii) added a short note that GNN implementations often place dropout and normalization after message passing and can differ materially

---

> ### Author Response · Authors · 2026-02-07
> **Reply to the Reviewer (3)**
>
> ## Survey Route
>
> >Systematize the prevention methods for over-smoothing (GNN) vs. rank collapse (Transformer) and over-squashing (GNN) vs. representational collapse (Transformer).
>
> We have expanded our analysis to explicitly survey and compare “depth-failure” mitigations. On the GNN side, we have grouped methods into: (a) normalization (e.g., PairNorm), and (b) skip/initial-residual/propagation control (e.g., GCNII, Jumping Knowledge Nets). On the Transformer side, we have connected these families to rank-collapse analyses of deep attention, and to block-level mitigation mechanisms (residual pathways, normalization, and scaling).
>
> To this end, we have included the following papers on:
> - GNN over-smoothing mitigation: Zhao & Akoglu 2019; Gasteiger et al. 2018; Chen et al. 2020; Xu et al. 2018; Li et al. 2020; Scholkemper et al 2025
> - Transformer rank collapse analysis: Dong et al. 2021; Noci et al. 2022; Wu et al. 2024; Chen et al. 2025; Saada et al. 2024; Joseph et al 2024; Alman et al 2025.
>
> To make the correspondence mechanistic (and not only conceptual), we have also included a Jacobian-based figure that illustrates how each architectural component (residuals, normalization/scaling, and mixing) reshapes the layer-wise Jacobian eigenspectrum, thereby explaining the gradient-stability properties shared by both systems.
>
> > Currently, your survey on over-squashing (GNN) vs. representational collapse (Transformer) consists on only two papers. Are there no works analysing over-squashing in GNNs?
>
> Thank you for pointing this out. We agree that, as written, our discussion of the Transformer-side “width” pathologies is anchored primarily around Barbero et al. (2024; 2025), which can give the impression that the literature is sparse. In our experience, many of the papers that cite these works do not directly study representational collapse / over-mixing as defined there (e.g., convergence of distinct last-token representations with increasing context under finite precision), but instead focus on closely related routing and extreme-token phenomena, such as attention sinks, sharpness/sparsity of attention, and gating mechanisms that prevent pathological concentration or diffusion of attention mass.
>
> For this precise reason, our intent in this paper is to encourage the community to develop a more systematic “width-regime” theory and empirical toolkit by importing ideas across fields (over-squashing in GNNs <=> collapse/over-mixing in Transformers). To better reflect the current landscape, in the revision, we have expanded this subsection to include additional representative works that, while not always phrasing their results explicitly in terms of representational collapse, provide complementary analysis and mitigation mechanisms for the same underlying issue of long-context information routing and mixing. Concretely, we have added discussion of training-free attention interventions (e.g., Han et al. 2025), geometric/optimization perspectives on sink formation  (e.g. Ruscio et al. 2025, Alcalde et al 2025), and gated attention mechanisms aimed at extreme-token phenomena (e.g. Bu et al. 2025, Fu et al 2026).
>
> >Section 4.3 discusses the design of a Transformer block. You need to add a discussion of a GNN block design.
>
> We have added a significant description of the canonical “deep GNN block” design and how it mirrors the Transformer block’s intent: maintain stable propagation and prevent collapse via residual/skip pathways, normalization, and controlled mixing strength. We will cite deep-GNN training recipes (e.g., DeeperGCN) and normalization/regularization mechanisms (PairNorm, etc.), and we will explicitly discuss similarities/differences in where normalization and dropout are applied in typical GNN implementations.
>
> ## Minor
>
> >Page 7. Figure 1 is a little misleading. First, it shows a tree, which is a very special case of graphs. I recommend presenting an arbitrary graph colouring a root node and 1-hop, 2-hop,... neighbourhoods (see Figure 1 in Zhang, et al. (2022) for the reference). Second, message passing GNNs and graph transformers propagate signal on a graph differently. Figure 1 is suitable for message-passing GNNs. I recommend pairing it side-by-side with a signal propagation picture for the transformer.
>
> We politely disagree that this figure is misleading: our intent is to show the computational tree rather than the underlying topological graph - similar to Figure 3 (right) in Arnaiz-Rodriguez & Errica (2025). This tree is unified between Transformers and GNNs, and “message passing” is equivalent from the perspective of this computational tree between these families of models, with the difference being the child nodes of the tree, which we make clear in the figure caption.
>
> > Page 8. The colour scheme on Figure 3 is hard to distinguish, especially when histograms are overlapping each other. Consider more contrastive colours.
>
> This is a good point, we have updated the color scheme used here.

---

> > ### Author Response · Authors · 2026-02-07
> > **Reply to the Reviewer (4)**
> >
> > > Page 9. "Transformers and GNNs have traditionally been viewed as distinct architectures" is generally wrong, because graph transformers are a thing and have to be mentioned in this paper.
> >
> > As detailed above, we have added more context about graph transformers.
> >
> > >Page 11. "In turn, modern LLMs ... impose a triangular (causal) mask for next-token generation. This, together with the softmax function, makes it row-stochastic." Consider rephrasing this bit for better readability. First, makes what stochastic? Second, if we assume "it" is "attention matrix", then softmax is solely responsible for row-stochasticity. The presence of causal masking has nothing to do with it. The attention matrix is row-stochastic even without the mask.
> >
> > Good point, we have fixed this.
> >
> > >Page 11. Causal transformers (like modern LLMs) are using mask to ensure the information flows only from the current input to future hidden states. Despite the masking is being mentioned on the same page, there is a factual error, where a causal graph is implied to be a fully connected graph.
> >
> > Good point, we will update this, although we have kept the citation from Joshi 2025 which makes this claim for completeness.
> >
> > >Page 12. Verify that Xiong et al. (2021) is the citation for pre-LN transformers. This paper does not seem to be about transformers.
> >
> > Good catch, we have fixed this. This got entangled with another Xiong entry in the bib file.
> >
> > ## References
> >
> > Alcalde et al. 2025: Attention's forward pass and Frank-Wolfe. https://arxiv.org/abs/2508.09628
> >
> > Alman, J., & Song, Z. (2025). Only large weights (and not skip connections) can prevent the perils of rank collapse. https://arxiv.org/abs/2505.16284
> >
> > Arnaiz-Rodriguez & Errica, 2025: Oversmoothing, Oversquashing, Heterophily, Long-Range, and more: Demystifying Common Beliefs in Graph Machine Learning. https://arxiv.org/abs/2505.15547
> >
> > Arroyo et al. 2025: On Vanishing Gradients, Over-Smoothing, and Over-Squashing in GNNs: Bridging Recurrent and Graph Learning. https://arxiv.org/abs/2502.10818
> >
> > Barbero et al. 2024: Transformers need glasses! Information over-squashing in language tasks. https://arxiv.org/abs/2406.04267
> >
> > Barbero et al. 2025: Why do LLMs attend to the first token? https://arxiv.org/abs/2504.02732
> >
> > Bu et al. 2025: Value-State Gated Attention for Mitigating Extreme-Token Phenomena in Transformers. https://arxiv.org/abs/2510.09017
> >
> > Chen et al. 2020: Simple and Deep Graph Convolutional Networks (GCNII). https://arxiv.org/abs/2007.02133
> >
> > Chen et al. 2025: Critical attention scaling in long-context transformers. https://arxiv.org/abs/2510.05554
> >
> > Choi et al. 2024: Graph Convolutions Enrich the Self-Attention in Transformers! https://arxiv.org/abs/2312.04234v5
> >
> > Dong et al. 2021: Attention is Not All You Need: Pure Attention Loses Rank Doubly Exponentially with Depth. https://arxiv.org/abs/2103.03404
> >
> > Fu, Zizhuo et al 2026: Attention Sink Forges Native MoE in Attention Layers: Sink-Aware Training to Address Head Collapse. https://arxiv.org/html/2602.01203v1
> >
> > Gasteiger et al. 2018: Predict then Propagate: Graph Neural Networks meet Personalized PageRank (APPNP). https://arxiv.org/abs/1810.05997
> >
> > Han et al. 2025: ZeroTuning: Unlocking the Initial Token's Power to Enhance Large Language Models Without Training. https://arxiv.org/abs/2505.11739
> >
> > Joseph at al 2024: Lambda-Skip Connections: the architectural component that prevents Rank Collapse. https://arxiv.org/abs/2410.10609
> >
> > Joshi, C. K. (2025). Transformers are graph neural networks. https://arxiv.org/abs/2506.22084.
> >
> > Kim & Ko 2024: Rethinking Attention Mechanisms in Vision Transformers with Graph Structures. https://www.mdpi.com/1424-8220/24/4/1111
> >
> > Li et al. 2020: DeeperGCN: All You Need to Train Deeper GCNs. https://arxiv.org/abs/2006.07739
> >
> > Luo, Y., Shi, L., & Wu, X. M. (2024). Classic gnns are strong baselines: Reassessing gnns for node classification. https://arxiv.org/abs/2406.08993
> >
> > Ma et al., 2023: Graph Inductive Biases in Transformers without Message Passing (GRIT). https://arxiv.org/abs/2305.17589
> >
> > Noci et al. 2022: Signal Propagation in Transformers: Theoretical Perspectives and the Role of Rank Collapse. https://arxiv.org/abs/2206.03126
> >
> > Rampášek et al., 2022: Recipe for a General, Powerful, Scalable Graph Transformer. (GraphGPS). https://arxiv.org/abs/2205.12454
> >
> > Ruscio et al. 2025: What are you sinking? A geometric approach on attention sink. https://arxiv.org/abs/2508.02546
> >
> > Saada et al. 2024: Mind the Gap: a Spectral Analysis of Rank Collapse and Signal Propagation in Attention Layers. https://arxiv.org/abs/2410.07799
> >
> > Scholkemper et al. 2024. Residual Connections and Normalization Can Provably Prevent Oversmoothing in GNNs. https://arxiv.org/abs/2406.02997
> >
> > Shehzad et al., 2024: Graph Transformers: A Survey. https://arxiv.org/abs/2407.09777
> >
> > Wu et al. 2024: On the Role of Attention Masks and LayerNorm in Transformers. https://arxiv.org/abs/2405.18781

---

> > > ### Comment · Reviewer_JEi5 · 2026-02-18
> > > **Response to the Authors**
> > >
> > > Thank you for the update. The revised manuscript looks much more solid. The majority of my comments was covered in the revision. In this comment, I only address some points specifically.
> > >
> > > > concepts as discussed in sections 2-4 generalize to the other modalities you mention (especially vision and graph)
> > >
> > > This is a fair point. I agree with it, and I would like to thank the authors for updating the motivation in the introduction. My comment is addressed.
> > >
> > > > highlight that our intent in drawing the “attention-as-learned-graph-operator”
> > >
> > > The point of "attention-as-learned-graph-operator not been widely adopted" is very debatable. Mostly because, again, graph transformers.
> > > At the same time, I acknowledge that there are very few papers adopting graph-related techniques for non-graph applications.
> > > In the manuscript, you focused on the latter, so I think this part of my comment was covered.
> > >
> > > > The inclusion of RNN background is deliberate
> > >
> > > Given a new list of adopted theorems in the revision, i believe a section about vanishing gradients is now useful. The useful material, however, is only first two paragraphs of section 2.1.3.
> > >
> > > In your work you are adopting some parts from Arroyo et al. (2025) and not presenting new theory. Arroyo et al. (2025) makes a connection between RNNs and GNNs, because it presents an "SMM-like" view of GNNs. Thus the background on RNNs and SMMs is relevant to Arroyo et al. (2025). In your manuscript you do a connection between Transformers and GNNs by arguing that some theory from Arroyo et al. (2025) devised for GNNs is applicable to Transformers. Therefore, the background on RNNs and SMMs is not relevant.
> > >
> > > **Additional changes**:
> > > Lemma 4.2.1 seems to be directly adopted from Arroyo et al. (2025). Please indicate this similarly to the others.

---

> > > > ### Author Response · Authors · 2026-02-22
> > > > **Responser to the Reviewer**
> > > >
> > > > Thank you for your response. We’re very happy to hear that the revisions have addressed the majority of your comments. We also appreciate you catching the missing attribution of Lemma 4.2.1, which has now been corrected. In addition, we removed the standalone RNN subsection and instead included a brief introductory paragraph to provide minimal historical context for the vanishing/exploding gradient problem, as it was first rigorously analyzed in the recurrent setting.

---

### Review · Reviewer_per9 · 2026-01-24

**Summary Of Contributions:**

**Summary:** The paper surveys the literature on optimization problems in graph neural networks and relates it to the attention mechanism in decoder-only transformer models, arguing that the architectural choices of LLMs may be explainable as attempting to prevent 'over-squashing' and 'over-smoothing' problems known to plague GNNs.

**Strengths:**
- The paper written in a relatively accessible manner and has a complete background for sequence models, self-attention, vanishing gradients, and graph neural networks.
- The paper serves as a strong review/survey of literature related to optimization and information propagation in graph-neural networks and sequence models.
- It introduces a potentially valuable connection between known issues with transformer architectures and known issues in the graph neural network and sequence modeling communities.

**Weaknesses:**
- The major weakness with the paper is the limited depth with which the new perspective is put to use to support the major claim that it provides a new 'understanding' of transformers.
	- The majority of the paper (up to page 9 & 1/2) is background or discussion that surveys prior work.
	- The main new contributions of the paper are presented in Section 4, but the formalism introduced early in the paper is not leveraged to make the claims convincing. Specifically:
		- Sentences such as: "We note that recent work (Qiu et al., 2025) also prevented the appearance of attention sink by incorporating gating into the attention mechanism, which can be seen as an alternative (and more direct) way of preventing over-mixing". This claims that their overmixing perspective applies to explain why the gating of (Qiu et al., 2025) is successful, but gives nothing to the reader in terms of how this perspective achieves this explanation.
		- Similarly, the idea of why sharp attention heads fix collapse in width is not explained at a mechanistic level, despite the introduction of the formal background earlier in the paper that would make this possible.
		- Ultimately, the quote in the conclusion is not fully justified: "This perspective offers a powerful framework for understanding how the spectral properties of attention matrices influence signal propagation and why architectural choices, like the Transformer block, are essential for robust learning." While this is likely true, the authors do not make a clear link between their perspective and the understanding they purport to provide.

- I do not agree with the sentence "While some of these works have attributed the success of their models to a superior inductive bias (e.g. through "physically-inspired methods"), it is more fundamentally tied to other optimization-related phenomena". First,  'other optimization-related phenomena' is non-specific enough to be of value to readers. Secondly, the distinction between physically inspired inductive biases and optimization related properties is an invalid distinction in my opinion. The physical systems these methods are based on have known stability properties that the authors are knowingly employing. I would argue it is disingenuous to claim that they are ignorant of the 'optimization-related' benefits of the physical systems they choose.

- On page 10: "Just as deep GNNs suffer from over-smoothing, where node representations converge to a uniform vector, naively stacking Transformer layers leads to rank collapse (Dong et al., 2021)". However, this is not what Dong 2021 proves, and does not align with the argument in section 4.3 -- Dong argues that stacking pure attention leads to rank loss, but skip connections and MLPs prevent this (just as the authors also argue later).

- Figure 5 is not intuitive, the meanings of the different colors are not explained, so it is not clear what is being represented.


**Minor:**
- Typo in footnote on page 4: "choice on nonlinearities"
- The Sutskever et al., 2013 citation on Page 4 seems out of place -- it is not about reservoir computing .
- "width" on page 6 and  "shallow” on page 9 have mixed quotation styles.
- The text on page 8 mislabels left vs. right for Figure 3: "This is illustrated in Figure 3 (right)," should say (left), and similarly for the next sentence.
- Page 12, you cite the wrong Xiong et al. paper. You cite 2021, but I believe you meant to cite the 2020 paper on Layer norm (https://arxiv.org/abs/2002.04745).
- A number of bibliography errors:
	- "The llama 3 herd of models" has an incorrect arxiv identifier.
	- "Diffwire" paper has a broken accent.
	-  Oversmoothing," oversquashing", -- extra quotes in title.

**Additional Comments:**

N/A

**Audience:**

Yes

**Audience Explanation:**

Rank collapse in decoder-only transformers is an active relevant topic in the literature that the current paper purports to provide a new lens and toolkit through which to study this phenomenon. The paper further offers a good review of the literature on oversquashing, oversmoothing, and vanishing gradients in recurrent neural networks.

**Claims And Evidence:**

Yes

**Claims Explanation:**

Since this is mainly a survey paper, the majority of the claims made are simply taken from prior work and thus true (except where mis-cited, as noted above e.g. for Dong et al.) -- justifying the answer 'yes'. The 'new' claims of the paper are also likely true, but the evidence for supporting them is left up to the reader to infer and therefore not clear or convincing (see major weakness above). I would therefore answer 'partial yes' if available.

**Requested Changes:**

- Addition of a concrete (mathematical) formalization of how the paper's perspective can explain the 'mitigation strategies' (or at least some of them).
- The colors in Figure 5 should be explained in more detail to make it interpretable.
- The noted minor issues (citations, typos, etc.).

---

> ### Author Response · Authors · 2026-02-07
> **Reply to the Reviewer (1)**
>
> We thank the reviewer for the thoughtful and detailed feedback. We are encouraged that the reviewer views the paper as accessible, relevant, and potentially valuable to the TMLR audience. Below, we address the main concerns point by point and outline the concrete changes we have made in the revision.
>
> > The main weakness is that, while the paper proposes a new perspective relating GNN pathologies to Transformer behavior, this perspective is not used with sufficient depth to explain mitigation strategies. Section 4 introduces relevant ideas, but the earlier formalism is not leveraged to make claims convincing. In particular:
> - Statements connecting gating to preventing over-mixing are not mechanistically justified.
> - The role of sharp attention heads in mitigating width-related collapse is asserted but not explained.
> - The conclusion claims a “powerful framework” without a clear chain of reasoning linking spectral properties to architectural choices.
> - A concrete mathematical formalization of how the perspective explains mitigation strategies is missing.
>
> We agree with this assessment and thank the reviewer for articulating it clearly. Our goal is primarily to bridge the GNN and Transformer communities by placing their optimization and information-propagation pathologies in a shared analytical framework; we acknowledge that, in the current draft, this perspective is stated more often than it is used.
>
> In the revision, we have strengthened Section 4 by explicitly leveraging the formal background to explain why mitigation strategies work, rather than merely noting that they do. Concretely, we have added a short mathematical formalization that links collapse phenomena to contractive dynamics of mixing operators, building on existing theory: for depth, we have adapted the Jacobian-product/contraction lens of Arroyo et al. (2025) to clarify how repeated attention-based mixing can drive rank collapse and vanishing gradients when the effective operator is strongly contractive; for width, we have connected over-mixing and representational collapse to sensitivity/bottleneck analyses as in Barbero et al. (2024, 2025), and used this to explain mechanistically how sharp attention and gating modify the relevant contraction/sensitivity behavior. Finally, we have revised the conclusion to more precisely reflect what is established by this unifying lens.
>
> > The distinction drawn between “physically inspired inductive biases” and “optimization-related phenomena” is not well defined and may be misleading. Physical systems are often chosen precisely because of their stability and optimization properties, and it is inappropriate to suggest otherwise.
>
> We agree with this critique. The wording in the current draft is imprecise and does not accurately reflect the relationship between inductive bias and optimization dynamics. We have revised this passage to remove the implied distinction and instead emphasize that many physically inspired architectures encode stability and favorable optimization properties by design.
>
> > Figure 5 is difficult to interpret because the meaning of the colors is not clearly explained.
>
> We agree. In the revision, we have expanded the caption of Figure 5 to explicitly explain what each color corresponds to, and we have also clarified in the main text which quantity is being visualized and how it connects to the surrounding discussion of width-related collapse, over-mixing, and attention behavior.
>
> > Several minor issues are noted, including typos, incorrect citations, and bibliography errors.
>
>
> We thank the reviewer for catching the minor issues and will correct them in the revision. In particular, we will fix the mis-citation and phrasing around Dong et al. (2021), clarifying the role of skip connections and MLP blocks in preventing rank loss in practice; we will correct the Xiong et al. citation to the appropriate LayerNorm reference; and we will resolve the remaining typos, formatting inconsistencies, and bibliography errors noted in the review.
>
> We included the citation to Sutskever et al. (2013) intentionally, as it is one of the earliest works to explicitly use echo-state–inspired spectral control ideas for RNN initialization, which is conceptually aligned with later developments such as LRUs. We have kept it for now, but we are also happy to remove the reference if the reviewer feels it remains out of place.

---

> ### Author Response · Authors · 2026-02-07
> **Reply to the Reviewer (2)**
>
> ## References
>
> Arroyo et al. 2025: On Vanishing Gradients, Over-Smoothing, and Over-Squashing in GNNs: Bridging Recurrent and Graph Learning. https://arxiv.org/abs/2502.10818
>
> Barbero et al. 2024: Transformers need glasses! Information over-squashing in language tasks. https://arxiv.org/abs/2406.04267
>
> Barbero et al. 2025: Why do LLMs attend to the first token? https://arxiv.org/abs/2504.02732

---

> > ### Comment · Reviewer_per9 · 2026-02-22
> >
> > I thank the authors for their response and taking the points raised in my review seriously. Their changes to phrasing of the noted weaknesses, citations, and figure captions, address my main concerns in that regard.
> >
> > Their revision of Section 4 is particularly appreciated. The inclusion of the additional propositions and theorems greatly strengthens the value of the manuscript as a unifying perspective and as a survey. The more detailed discussion of shared mitigation strategies also address my previous concerns regarding the applicability of the perspective. As the authors note, they now clarify how 'common stabilizers intervene' in existing failure modes in a unified way.

---

### Decision · Action_Editor_3YhN · 2026-03-24

**Recommendation:** Accept as is

**Audience:**

Yes

**Audience Explanation:**

The paper presents a valuable unifying perspective that has the potential to bring together diverse communities to exchange ideas about how to address common challenges.

The paper is very likely to be of interest to some of TMLR's audience.

**Claims And Evidence:**

Yes

**Claims Explanation:**

The paper makes the following claims:

(1)	It connects decoder self-attention and message-passing GNN updates, showing how they can both be interpreted as performing mixing followed by feature updates.

(2)	It links over-smoothing and over-squashing from the GNN literature to rank collapse and long-context degradation.

(3)	It shows how there are common themes in the mitigation strategies adopted for GNNs and transformers;

These claims are supported by the presented theory, methodology, and discussion.